# Medial orbitofrontal inactivation does not affect economic choice

**Matthew PH Gardner[1]\*, Jessica C Conroy[1], Clay V Styer[1], Timothy Huynh[1], Leslie R Whitaker[1], Geoffrey Schoenbaum[1,2,3,4]\***

[1]NIDA Intramural Research Program, Baltimore, United States; [2]Department of Anatomy & Neurobiology, University of Maryland School of Medicine, Baltimore, United States; [3]Solomon H. Snyder Department of Neuroscience, The Johns Hopkins University, Baltimore, United States; [4]Department of Psychiatry, University of Maryland School of Medicine, Baltimore, United States

**Abstract** How are decisions made between different goods? One theory spanning several fields of neuroscience proposes that their values are distilled to a single common neural currency, the calculation of which allows for rational decisions. The orbitofrontal cortex (OFC) is thought to play a critical role in this process, based on the presence of neural correlates of economic value in lateral OFC in monkeys and medial OFC in humans. We previously inactivated lateral OFC in rats without affecting economic choice behavior. Here we inactivated medial OFC in the same task, again without effect. Behavior in the same rats was disrupted by inactivation during progressive ratio responding previously shown to depend on medial OFC, demonstrating the efficacy of the inactivation. These results indicate that medial OFC is not necessary for economic choice, bolstering the proposal that classic economic choice is likely mediated by multiple, overlapping neural circuits.
DOI: https://doi.org/10.7554/eLife.38963.001

\*For correspondence:
matthew.gardner2@nih.gov
(MPHG);
geoffrey.schoenbaum@nih.gov
(GS)

**Competing interests:** The authors declare that no competing interests exist.

## Introduction

Decision-making often requires us to make choices between biologically relevant goods that differ across many dimensions, including some whose value may be largely contextual (i.e. I value water most when I am thirsty) or even highly subjective (i.e. I like blue). How the brain accomplishes such comparisons, maintaining the relative values of disparate goods across time, while adjusting to reflect new situations and even contingencies, is a fundamental question in cognitive neuroscience. Neuroeconomics holds that a major step in this process is the compression of the relevant features defining the different goods into a single dimension of value, so that disparate goods can be compared on a universal scale (*Levy and Glimcher, 2012*; *Padoa-Schioppa, 2011*; *Padoa-Schioppa et al., 2006*). Single-unit and fMRI studies have typically assigned this process to the orbitofrontal cortex (OFC) based on the demonstration that neural activity there – identified either by single-unit or BOLD response – tracks value on this universal scale during economic choice (*Levy and Glimcher, 2011*; *McGinty et al., 2016*; *Padoa-Schioppa, 2009*; *Padoa-Schioppa and Assad, 2006*; *Padoa-Schioppa and Assad, 2008*; *Plassmann et al., 2010*; *Plassmann et al., 2007*; *Rich and Wallis, 2016*; *Tremblay and Schultz, 1999*; *Xie and Padoa-Schioppa, 2016*). Yet these correlates are not ubiquitous; often single-units in orbitofrontal cortex do not seem to encode value independent of other information (*Blanchard et al., 2015*; *Kennerley et al., 2011*; *Kennerley and Wallis, 2009*; *McDannald et al., 2014*; *Wikenheiser et al., 2017*). Further, causal evidence to support a critical role for the OFC in economic choice is sparse, owing to the difficulty in doing such experiments in humans or monkeys, where economic choice is typically studied.

To address this, we recently duplicated in rats the economic choice task used in monkeys to isolate key neural correlates of economic value in the lateral OFC (*Padoa-Schioppa and Assad, 2006*). Although rats' behavior in this task exhibited all the core features of economic choice (*Levy and Glimcher, 2012*; *Padoa-Schioppa, 2011*; *Padoa-Schioppa et al., 2006*), it was entirely insensitive to optogenetic inactivation of the lateral OFC, inactivation which was sufficient to disrupt devaluation-sensitive changes in Pavlovian responding (*Gardner, 2017*. Indeed recent work suggests that lateral OFC is not generally necessary for established choice behavior (*Miller et al., 2018*).

While we interpreted our result as consistent with the parallel support of economic choice by a number of neural circuits, an alternative explanation is that the lateral OFC is simply the wrong part of OFC. The OFC is increasingly recognized as consisting of several subregions (*Heilbronner et al., 2016*; *Izquierdo, 2017*; *Murphy and Deutch, 2018*; *Rudebeck and Murray, 2011a*; *Walton et al., 2011*), and medial OFC is implicated in signaling common currency value by human imaging studies (*Levy and Glimcher, 2011*; *Plassmann et al., 2010*; *Plassmann et al., 2007*) and in mediating value-guided behaviors in rats (*Bradfield et al., 2015*; *Münster and Hauber, 2017*), monkeys (*Noonan et al., 2010*) and humans (*Camille et al., 2011*; *Noonan et al., 2017*). Here we tested whether the medial region might be the critical site in OFC responsible for calculating economic value by optogenetically inactivating this region in rats engaged in the aforementioned task.

## Results

To test whether medial OFC is critical for economic choice behavior we inactivated it while rats performed an economic choice task, developed previously (*Gardner et al., 2017*) to mimic procedures used in monkeys (*Padoa-Schioppa and Assad, 2006*). In this task, rats learned to choose between different numbers of flavored food pellets, providing the opportunity to precisely measure each rats' relative preferences between different goods. On each trial within the task, an offer was displayed on a pair of 3.5' touchscreens situated on either side of a central nosepoke. Each offer consisted of two symbols that conveyed two pieces of information: 1) the pellet flavors in the offer (banana, bacon, grain, chocolate, grape or cellulose), indicated by the shape of each symbol, and 2) the number of pellets of each flavor in the offer (1,2,3,4,6 and sometimes 8), indicated by the segmentations of each symbol (*Figure 1A*). Rats were required to nosepoke for 1s while the offer was displayed, after which they were able to make a choice by touching the screen with the preferred option. The chosen pellets were subsequently delivered in a food magazine situated on the opposite wall of the operant chamber. Each session consisted of 11 offers in which two food pellet flavors were presented in different ratios. Rats learned six visual cue –pellet pairings, and experienced the 15 possible pairings of the six pellet-types across training and testing.

We have previously demonstrated that within this task rats display rational economic choice behavior (*Gardner et al., 2017*). Within individual sessions, rats showed a clear change in preference for the two pellet-types across the range of offers given. The offer for which rats showed no preference for the two offer-types, referred to as the indifference point (IP), was used as an estimate of the relative valuation of the two pellet flavors. Notably, indifference points between particular pairs of food pellets for individual rats ranged widely from 1:1 to greater than 8:1, suggesting that the rats were choosing based on internal subjective values (only IPs <= 6 were included in analyses). Further these indifference points were stable across sessions and exhibited transitivity across offer pairs with shared constituents, a property often taken as an indicator of rational decision making.

Given that medial OFC has been suggested to represent economic value, we expected that these hallmark features of economic decision making might be disrupted if it were taken offline in the critical choice phase of each trial. To that end, we prepared rats so that we could transiently inhibit neural activity in medial OFC using the optogenetic inhibitory protein, halorhodopsin (NpHR3.0). Rats (n = 6) were first trained on the task before undergoing surgery. AAV-CaMKIIa-eNpHR3.0-eYFP was then infused into medial OFC bilaterally, and fiberoptic probes were implanted just above each injection site (*Figure 1C*). Following surgery and a 2 – 3 week postoperative recovery period, rats were retrained to pre-surgical performance levels (~5 weeks, see Materials and methods), then we began sessions in which we tested the effects of inactivating medial OFC.

This phase of the experiment, lasting ~8 weeks, was designed as follows: rats experienced three separate pellet-pairs for three consecutive days for a total of 9 test sessions (*Figure 1B*). Rats experienced each pair in an initial warm up session that was followed by two test sessions in which they

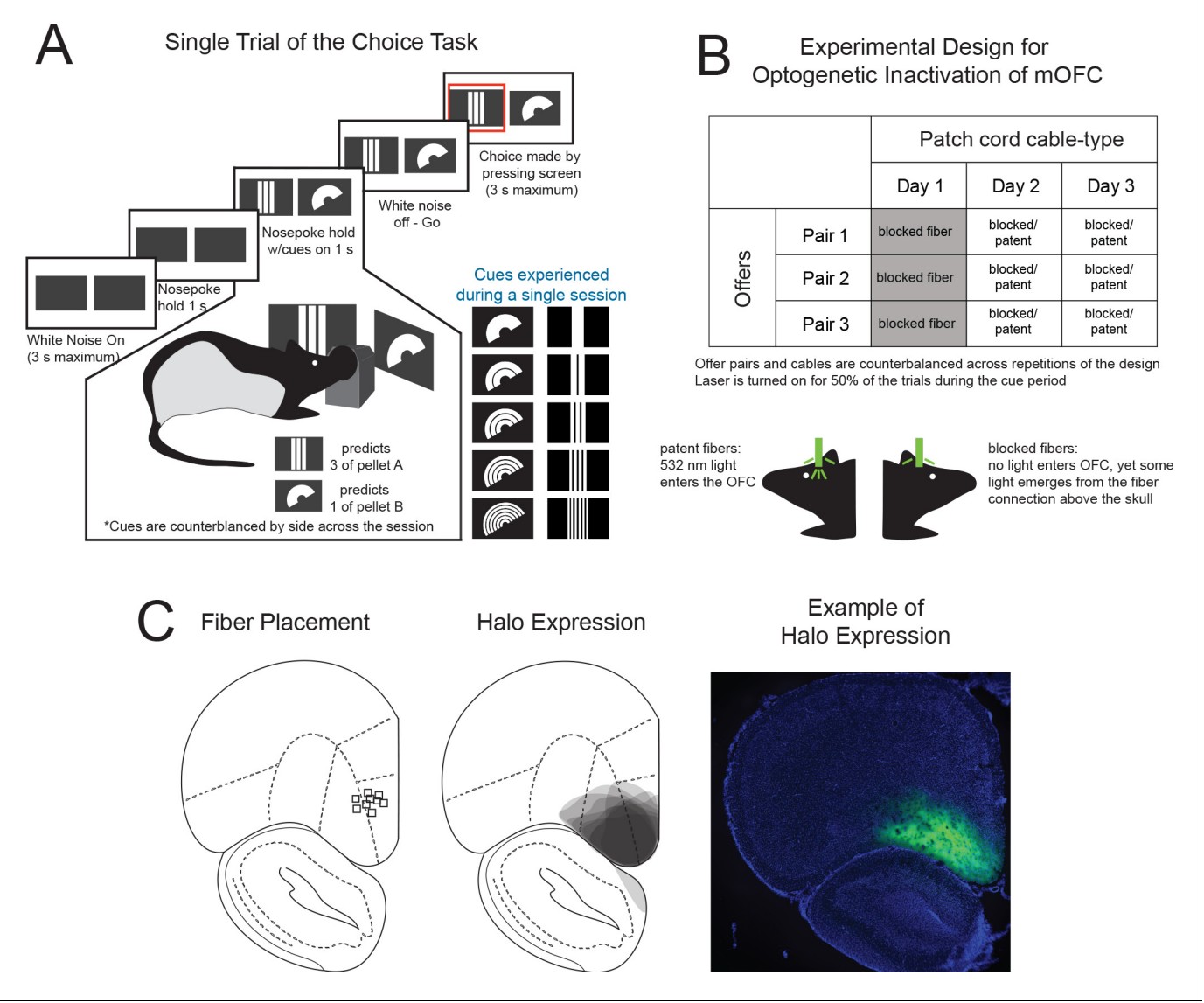

**Figure 1.** Inactivation of medial OFC on Economic Choice Behavior in Rats during the Cue Period. (**A**) Illustrative example of a single trial on the choice task. White noise cues the availability to initiate a trial by nosepoking within a central port. Images representing the current offer emerge on both screens after a 1s hold. Rats must then continue to hold for 1s, while visualizing the offers, before the white noise is turned off indicating that the animal can then touch one of the screens in order to make a choice. The shapes of the two displayed symbols signify the goods being offered (i.e. vertical bars and crescents), whereas the number of segmentations of the symbols indicate the number of pellets available for each good. For each session, rats experience one pair of the 10 possible pairs of learned associations. (**B**) Design for the inactivation experiment. (**C**) Histological verification of viral expression (middle) and fiber placement (left) for each of the rats at ~4.7 mm anterior of bregma. Example of NpHR3.0-eYFP expression (green) and DAPI (blue) (right).

DOI: https://doi.org/10.7554/eLife.38963.002

were connected to either a dummy cable, which was blocked at the ferrule to prevent light delivery to the brain, or a patent light-transmitting cable, which was unblocked at the ferrule allowing light to enter the brain. Cable type was counterbalanced across days. Each session was also separated into trials in which the laser was either on or off. During laser-on trials, green light (16 – 18 mW, 532 nm) was delivered to the fiberoptic cables. Laser onset co-occurred with the onset of the white noise that signaled that a trial could be initiated. The laser was terminated when the rat contacted one of the touch screens, and termination included a period of 300 ms during which the power was reduced linearly, a protocol which has been shown to minimize or prevent rebound excitation (***Chuong et al.,***

2014). Note, this design was identical to our previous experiment in which we inactivated the lateral OFC, except that we did not include a group injected with virus lacking the coding region of the hal-orhodopsin chloride pump. Because the behavior of the eYFP group in the prior study was identical to that of the experimental group and given the presence of both blocked-fiber control sessions and no-laser control trials, we deemed this third, between-subject control to be superfluous and hence unnecessary for the current experiment.

An initial examination of data from example sessions (*Figure 2A*, blocked and patent cable sessions from consecutive days on the same pellet pair) revealed very little difference in choice behavior between the laser-on and –off trials, and the averaged choice behavior (*Figure 2B*) supported this assessment. To look more closely, we quantified the effect of medial OFC inactivation on three separate primary behavioral measures: (1) the indifference point or relative preferences between pellet-types; (2) the steepness of the preference curves; and (3) transitivity or the consistency of preferences across multiple pellet-pairs experienced over a series of days.

If the medial OFC is necessary for the assignment of value to particular goods or outcomes with the quantity of those goods modulating the decision elsewhere, one might expect a shift of the IP towards a 1:1 preference. To determine whether this occurred with medial OFC inhibition, we ran a two-factor repeated measures ANOVA on 50 pairs of sessions with factors Laser (laser-on/-off) and Fiber (blocked/patent). This analysis revealed no significant main effects of either Laser ($F_{(1,147)}$ = 1.31, p=0.25) or Fiber ($F_{(1,147)}$ = 1.03, p=0.31) nor any significant effect of the critical interaction term Laser*Fiber ($F_{(1,147)}$ = 0.75, p=0.39). In addition, at the reviewers' request, we also compared the current groups to eYFP controls, identically trained and tested in a prior experiment, and there were no significant differences (*Figure 3—figure supplement 1*, see *Supplementary file 1* for ANOVA results)(*Gardner et al., 2017*). We also ran post-hoc simple effects comparing levels across each factor of the design. None of the four comparisons reached significance (paired t-test, n = 50, p>0.14, β <1e-10 for a shift to 1:1); this included the primary comparison of interest, laser-on vs laser-off trials in the patent cable condition (p=0.90). The indifference point increased by only 0.3% during the laser-on trials (mean IP for the patent fiber and laser-off trials: 1.649, mean IP for laser-on trials: 1.65), an effect which is well within the confidence interval (CI: [1.56 1.74], which corresponds ±10.6% shift of the IP). Importantly, our test would have found about a one-tenth change in the average preference significant.

Since rats do not have strong preferences between some of the pairs of food pellets, it's possible that mild preferences close to 1:1 might mask a significant IP shift towards a 1:1 ratio with medial OFC inactivation. To test this possibility, we removed sessions within a conservative 1.45:1 preference based on an empirical distribution of variance (see Materials and Methods, 19 session pairs were excluded) and reran the two-way ANOVA. Despite our focus on sessions with strong preferences (mean IP for all conditions: 2.07), this analysis still found neither significant main effects (Fiber: $F_{(1,90)}$ = 1.57, p=0.21; Laser: $F_{(1,90)}$ = 1.19, p=0.28) nor any interaction ($F_{(1,90)}$ = 0.87, p=0.35). We again tested the simple effects of the design of which none were significant (4 paired t-tests, n = 31, p>0.15, β <1e-10 for a shift to 1:1) including the primary comparison, laser-on vs laser-off trials in the patent cable condition, (p=0.90). The mean IP increased 0.40% from 2.03 (laser-off) to 2.04 (laser-on) with a 95% confidence interval of [1.83 2.25] or ±17.2% of the mean laser-off IP. The exclusion of sessions with IPs close to the 1:1 can be visualized in *Figure 3D* in which we progressively removed the closest IP to 1:1 from the dataset up to 90% of the sessions. The vertical line represents the threshold in which we performed the above ANOVA with 31 pairs of sessions.

Because the full course of inactivation spanned ~8 weeks, we wanted to be sure there was no effect of time influencing our measurements of inactivation of mOFC on the indifference point. Although beginning 7–8 weeks after the initial viral injections, which should allow ample time for expression of the light-sensitive chloride pump, the effect of inactivation might have evolved over the course of the experiment. To test this we split the data into thirds, coinciding with the design for the test of transitivity. *Figure 2—figure supplement 1* shows the average behavior for each of the three blocks of sessions. It is apparent from visual inspection of the plots that there was no effect of inactivation at any stage of the inactivation. A three-way ANOVA with factors Laser, Fiber and Time revealed no significant effect of the critical three-way interaction of Laser*Fiber*Time ($F_{(2,141)}$ = 0.08, p=0.92), nor any of the two-way interactions (Laser*Time: $F_{(2,141)}$ = 1.58, p=0.21; Fiber*Time $F_{(2,141)}$ = 0.35, p=0.70; Laser*Fiber ($F_{1,141)}$=0.73, p=0.40). There was a main effect of Time (F

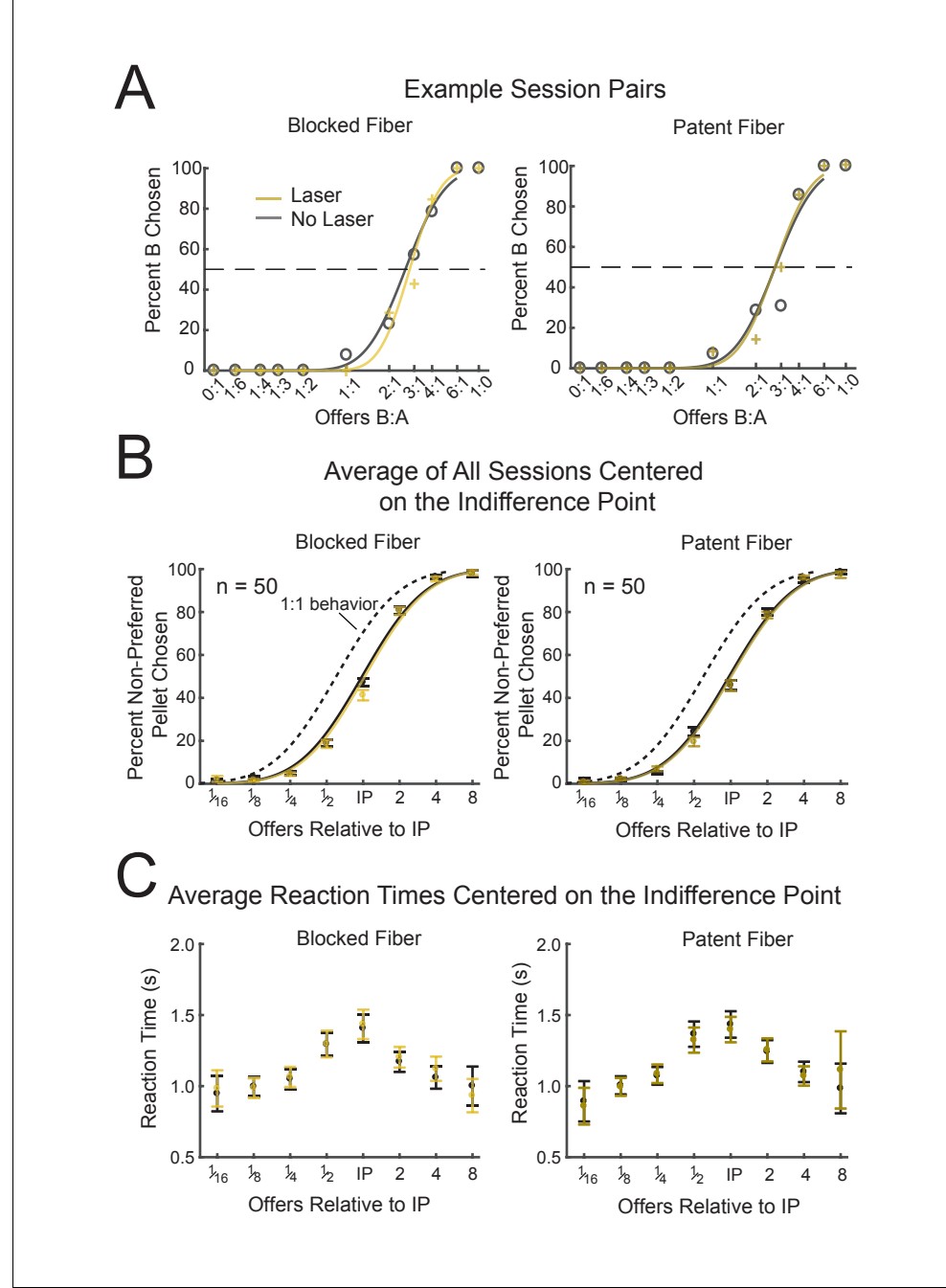

**Figure 2.** Effects of Optogenetic Inhibition of Medial OFC on Economic Choice. (**A**) Typical examples of observed behavior on the choice task during sessions with inactivation of medial OFC using halorhodopsin (patent fiber, right panel) or during control sessions (blocked fiber, left panel). The trials were equally split with the laser either on (yellow) or off (grey). (**B**) Average behavior of all sessions on the choice task for blocked (n = 50, left) and patent fibers (n = 50, right) and for laser-on (yellow curves) and laser-off (grey curves) trials. To show the mean behavior for all sessions, the chosen offers from each session were first realigned to the IP of that session before being averaged. Note that the two curves, laser-on and laser-off, are overlapping one another in most panels. The dark grey dotted line shows the average behavior expected if all sessions shifted to an IP of 1:1. Error bars: SEM. (**C**) Average reaction times for blocked (left) and patent fibers (right) as well as for laser-on (yellow) or laser-off (dark grey) trials. Offers are binned relative to the indifference point of a session (X-axis) as in (**B**) Average reaction times are in seconds (Y-axis, average of all sessions shown in (**B**). Error bars: SEM. See *Figure 2—figure supplement 1* for breakdown of average behavior over the course of the experiment.
DOI: https://doi.org/10.7554/eLife.38963.003

*Figure 2 continued on next page*

*Figure 2 continued*

The following figure supplement is available for figure 2:

**Figure supplement 1.** Average Effects of Medial Orbitofrontal Inactivation on the Economic Choice Divided into Three Epochs.

DOI: https://doi.org/10.7554/eLife.38963.004

(1,141) = 5.97, p=5.0e-3), which is likely due to the variations in the preferences across the three pellets used for each of the three time blocks.

Effects of time could also be manifest within each session. To determine whether an effect of inactivation was apparent at the beginning of a behavioral session, we included the minimum number of trials for each offer for which we could determine a reasonable estimate of the IP (n = 4 for each of the 11 offers, for both laser-on and –off trial-types, 88 trials total) which was on average, the

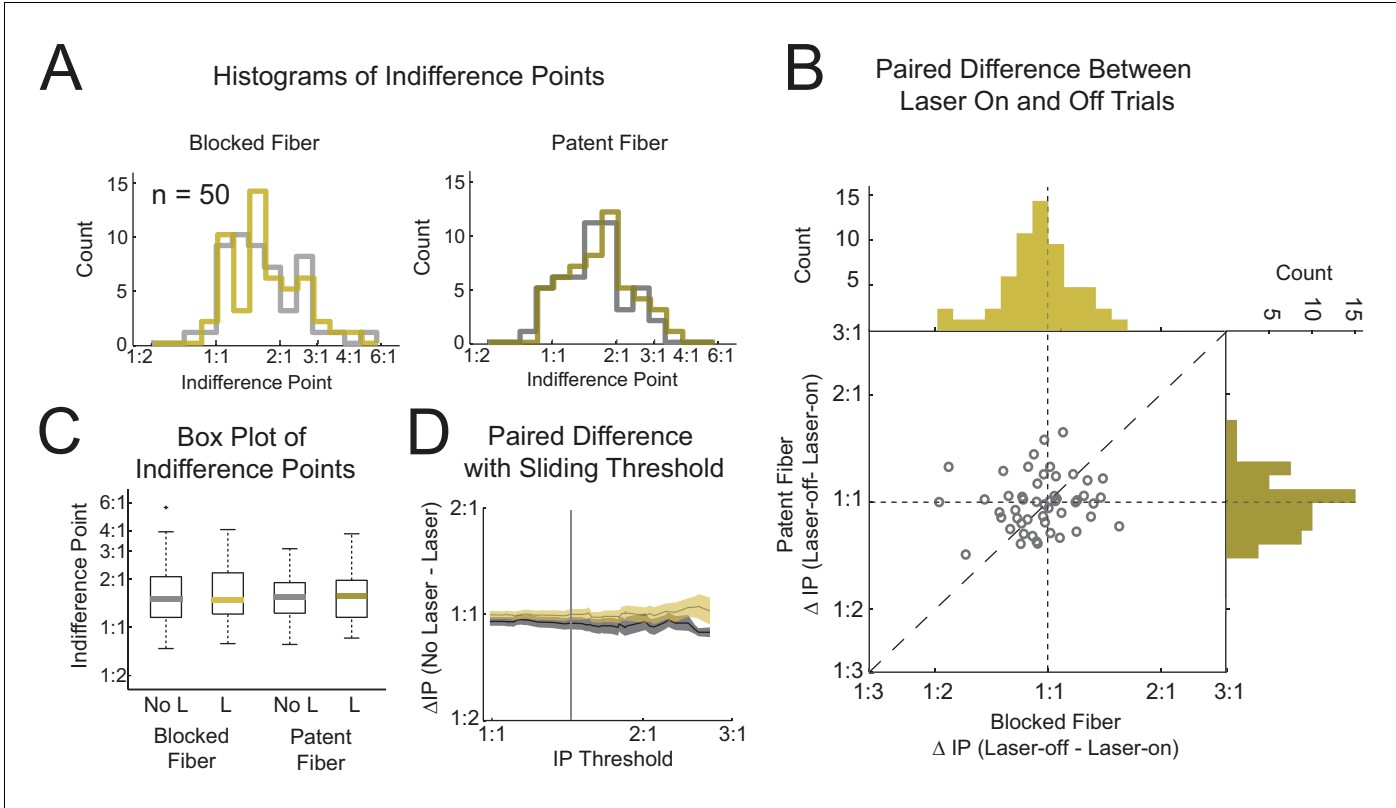

**Figure 3.** Optogenetic Inhibition of Medial OFC during the Cue Period Does Not Affect the Indifference Point. Histograms of the indifference points (IPs) from 50 sessions are shown for the blocked and patent fibers (A left/right), split by laser-on trials (yellow) and laser-off trials (grey). (B) A scatterplot showing the difference between same session laser-on and laser-off trials for paired blocked and patent fiber sessions (x- and y-axis respectively) with the corresponding histograms on each axis. (C) Boxplots of IPs for each treatment corresponding to the histograms in (A). To be sure that shifts in the indifference point were not masked by sessions with IPs near 1:1, sessions close to 1:1 were progressively removed using a sliding threshold (D). The average difference in IP for laser-on and -off trials were calculated only using sessions with IPs greater than the IP threshold (x-axis) for the blocked (grey) and patent (yellow) fiber sessions. The grey vertical line in (D) is the IP threshold corresponding to an IP significantly shifted from 1:1 (equal preference) as determined by a bootstrap method (see Materials and methods). All axes representing indifference points are plotted in log scale. See *Figure 3—figure supplement 1* for comparison with virus control group.

DOI: https://doi.org/10.7554/eLife.38963.005

The following source data and figure supplement are available for figure 3:

**Source data 1.** Text File Containing the Source Data for Figure 3.

DOI: https://doi.org/10.7554/eLife.38963.007

**Figure supplement 1.** Comparison of the Effects of Optogenetic Inhibition of Medial OFC on the Indifference Point with a Virus Control Group.

DOI: https://doi.org/10.7554/eLife.38963.006

first 29.9% of the session. We then reran the two-way ANOVA test on this early session behavior. Neither the critical interaction effect of Laser*Fiber nor the main effects displayed a significant effect of inactivation (see *Supplementary file 2* for ANOVA results).

We next tested whether inhibition of medial OFC might be increasing the variance or uncertainty of the value assessment on individual trials. This effect in isolation would affect the slope of the fit without impacting the IP. Histograms and boxplots of the inverse slope, or the variance estimate of the probit fit, are shown in *Figure 4A–C*.

A repeated measures two-way ANOVA on these measures revealed no significant main effect of either Laser (F(1,147) = 0.69, p=0.41) or Fiber (F(1,147) = 0.11, p=0.74) nor any significant effect of the critical interaction term Laser*Fiber (F(1,147) = 0.26, p=0.61). In addition, at the reviewers' request, we also compared the current groups to eYFP controls, identically trained and tested in a prior experiment, and there were no significant differences (*Figure 4—figure supplement 1*, see *Supplementary file 1* for ANOVA results)(*Gardner et al., 2017*). Simple effects within and across factors also showed no significant differences (4 paired t-tests, n = 50, p>=0.29). The variance

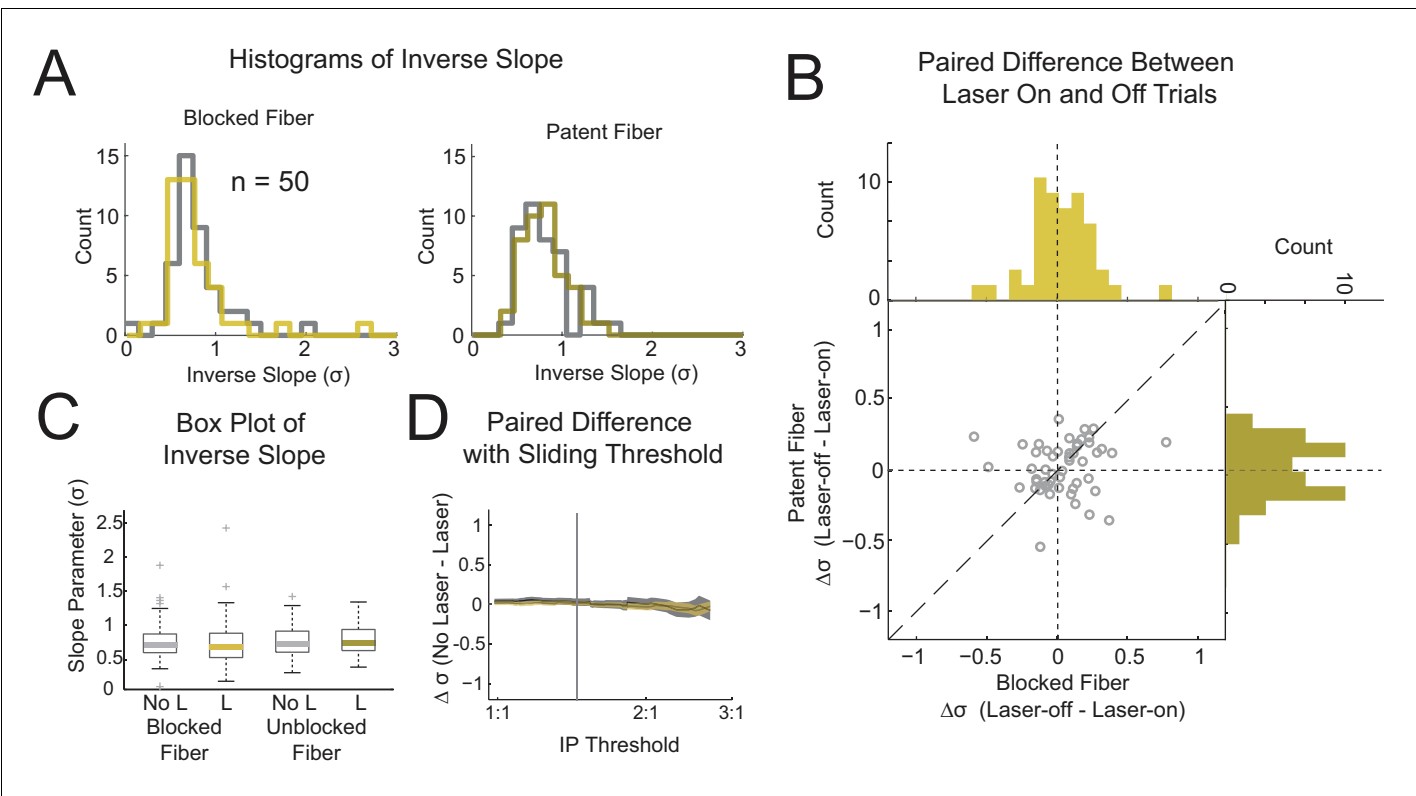

**Figure 4.** Optogenetic inhibition of medial OFC during the cue period does not affect the inverse slope. Histograms of the inverse slope (σ, probit regression) from 50 sessions are shown for the blocked and patent fibers (**A** left/right), split by laser-on trials (yellow) and laser-off trials (grey). (**B**) A scatterplot showing the difference between same session laser-on and laser-off trials for paired blocked and patent fiber sessions (x- and y-axis respectively) with the corresponding histograms on each axis. (**C**) Boxplots of the inverse slope for each treatment corresponding to the histograms in (**A**). (**D**) The average difference in inverse slope for laser-on and -off trials calculated only using sessions with IPs greater than the IP threshold (x-axis) for blocked (grey) and patent (yellow) fibers. For this analysis sessions close to 1:1 were progressively removed to up to 90% of the total sessions analyzed. The grey vertical line in (**D**) is the IP threshold corresponding to an IP significantly shifted from 1:1 (equal preference) as determined by a bootstrap method (see methods). All axes representing indifference points are plotted in log scale. See *Figure 4—figure supplement* for comparison with virus control group.

DOI: https://doi.org/10.7554/eLife.38963.008

The following source data and figure supplement are available for figure 4:

**Source data 1.** Text File Containing the Source Data for Figure 4.
DOI: https://doi.org/10.7554/eLife.38963.010

**Figure supplement 1.** Comparison of the Effects of Optogenetic Inhibition of Medial OFC on the Slope with a Virus Control Group.
DOI: https://doi.org/10.7554/eLife.38963.009

estimate decreased 1.0% for the laser-on trials ($\sigma$=0.84) as compared to laser-off trials ($\sigma$=0.83) in the critical patent fiber condition. A change well within the 95% confidence interval [0.76 0.91]. We again tested whether the slope might be more affected by inactivation during sessions in which the IP is reasonably distant from equal or 1:1 preference (*Figure 4D*). We used the same empirical threshold, 1.45:1, that we used for testing the IPs resulting in 19 fewer session pairs entering into the analysis. We repeated the ANOVA and again found no significant effects (Laser: F(1,90) = 0.19, p=0.66; Fiber: F(1,90) = 0.74, p=0.39; Laser*Fiber: F(1,90) < 0.01 , p=0.98), and testing of the simple effects across conditions showed no effects as well (4 paired t-tests, n = 31, p>0.55). For the critical comparison in the patent fiber condition the change in the variance estimate between laser-off ($\sigma$ = 0.82) and laser-on ($\sigma$ = 0.80) was well within the CI: [0.47 0.90] and was even in the direction of a steeper slope rather than the predicted flattening of the slope.

We also tested effects of time on the slope paralleling our analysis of the IP - we split the dataset into thirds, coinciding with the design for the test of transitivity (*Figure 2—figure supplement 1*). Again it is easily apparent from the figure that there was no emergent effect of inactivation over time. A three-way ANOVA with factors Laser, Fiber and Time revealed no significant effect of the critical three-way interaction of Laser*Fiber*Time (F(2,141) = 0.06, p=0.94), nor any of the two-way interactions (Laser*Time: F(2,141) = 0.04, p=0.96; Fiber*Time F(2,141) = 2.11, p=0.12; Laser*Fiber (F1,141)=0.27, p=0.61). Unlike the IP, there was no main effect of Time (F(1,141) = 0.44, p=0.64), further indicating this effect was due to variations in the preferences of the pellets used across the time blocks.

Finally, we looked at whether the slope was affected by inactivation just at the beginning of the session. As explained for the IP analysis, we narrowed our data to the first 4 trials of each offer (on average the first 29.9% of each session) and reran the two-way ANOVA test on this early session behavior. The critical interaction effect of Laser*Fiber was not significant. The main effect of Laser was significant. Further analysis revealed a significant difference in the simple effects between the laser-on and –off trials in the blocked fiber condition, but not in the patent fiber condition (see *Supplementary file 2* for ANOVA results).

After determining that medial OFC inhibition did not result in any observable within-session differences, we next examined whether medial OFC inhibition affected the stability and thereby transitivity of the rats' economic choice behavior across sessions. Rats show transitive behavior on our task when choosing between multiple pellet-types across days. This implies that rats have a consistent and relatively stable value space onto which they map different goods in our task. Medial OFC inactivation could disrupt this mapping, resulting in stable behavior during single sessions but disrupted transitive behavior across sessions. To assess this, we plotted transitivity from 15 sets of 3 pellet-pairs presented across 6 sessions (*Figure 5B*). An example of the behavior for one transitivity measure, spanning 6 sessions, is shown in *Figure 5A*. Each point on the scatterplots represents the consistency of pellet preferences from a group of three sessions. The greatest of the three preferences is plotted on the y-axis, $IP_{A:C}$, and the product of the other two pairs are plotted on the x-axis, $IP_{A:B} \times IP_{B:C}$. Stability of food preferences should be observed as points falling close to the identity line such that $IP_{A:C} \approx IP_{A:B} \times IP_{B:C}$ (*Gardner et al., 2017*); *Padoa-Schioppa and Assad, 2008*). Laser-on and laser-off trials from the same three-session pairs are connected by lines on the plot in order to visualize the effect of medial OFC inactivation on the same sessions. A comparison of these points for laser-on versus laser-off trials in blocked versus patent sessions demonstrates that the rats' behavior was again generally transitive, with only small, seemingly random fluctuations between the laser-on and laser-off trial types. Critically the points and these small fluctuations were unaffected by whether or not the fiber was patent to allow inactivation. A table showing the IPs from all transitivity session sets is shown in *Supplementary file 3*. To confirm this, we performed a repeated-measures, two-way ANOVA comparing the single transitivity values; this analysis found neither main effects nor any interaction (Laser: F(1,42) = 0.54, p=0.47; Fiber: F(1,42) = 3.21, p=0.08; Laser*Fiber: F(1,42) = 0.54, p=0.47).

The previous analysis demonstrated that the transitive nature of the behavior did not change during medial OFC inhibition. We also wanted to test whether the paired distances of the transitivity points (laser-on vs laser-off trials), when plotted in the two dimensional space of the $\log IP_{A:B} + \log IP_{B:C}$ (x-axis in *Figure 5B*) and the $\log IP_{A:C}$ (y-axis in *Figure 5B*), increased with inhibition of medial OFC, as might happen if inhibition were increasing the variability of valuation in the goods space. These distances are plotted as the black lines connecting the paired dots in *Figure 5B*

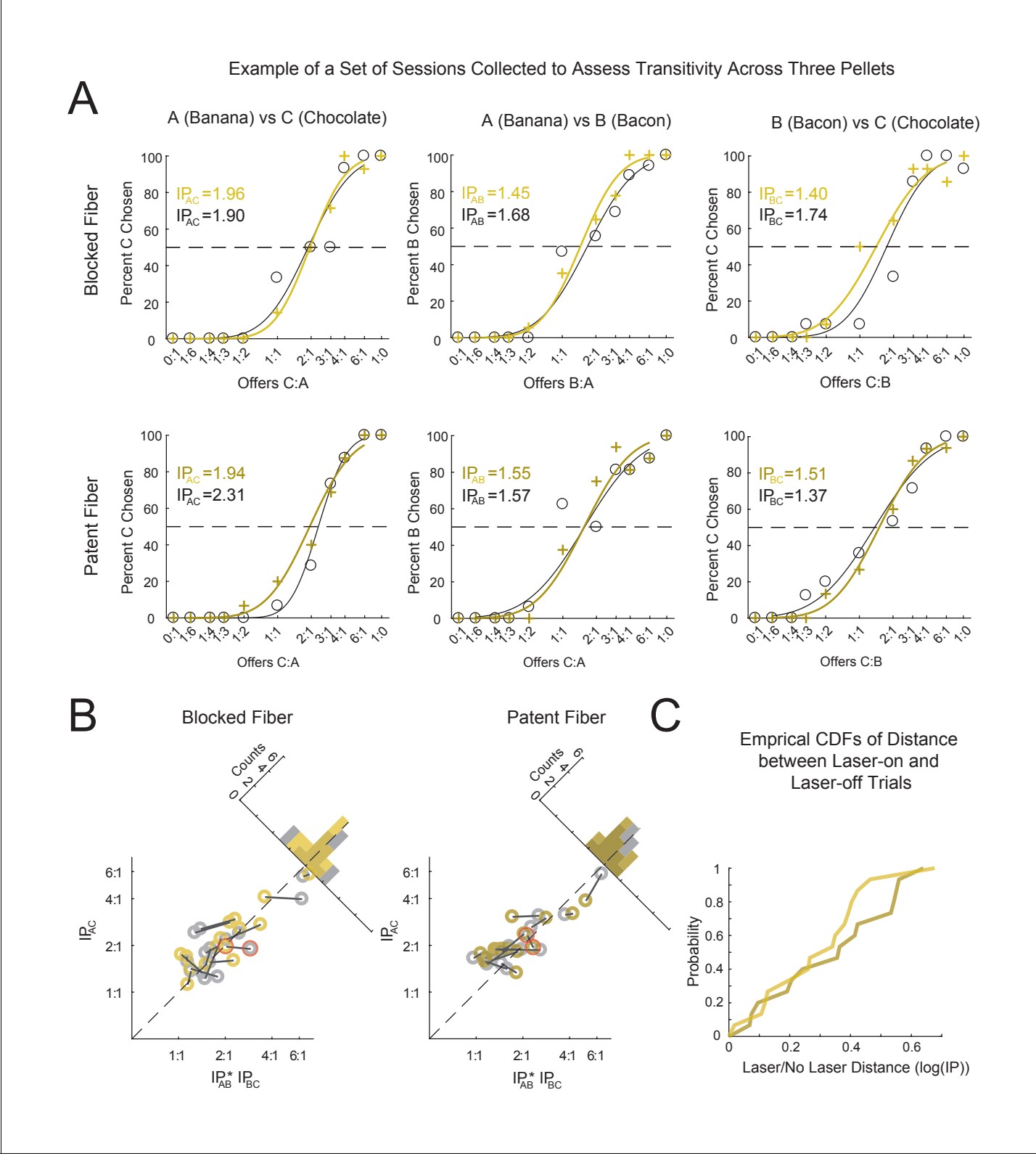

**Figure 5.** Transitivity Is Unaffected by medial OFC Inactivation. (A) Example of economic choice behavior from six sessions comprising one experimental unit of the test for transitivity. Sessions in each column show choice behavior between each pairing of three pellets referred to as (A, B, and C) based on decreasing order of preference (left: pair with the greatest IP - A vs C; middle: A vs B; right: B vs C). Top and bottom row show consecutive sessions, counterbalanced for order, in which the blocked or patent fiber, respectively, was used. Behavior is shown for both laser-on

*Figure 5 continued on next page*

*Figure 5 continued*

(yellow) and laser-off (trials) as well as the respective indifference point for each condition. (**B**), Transitivity plots are shown for the blocked (left) and patent (right) fiber types. Transitivity measures represent the consistency of food pellet preferences across six sessions in which animals experienced each of the possible pairs of three pellets (A–C,A–B,B–C) for both fiber-type conditions. Stable and consistent preferences should be in accordance with the relation: $IP_{A:C} \approx IP_{A:B} \times IP_{B:C}$ which can be visualized in the scatterplots as points falling close to the identity line. The Y- and X- axes show the IP, in log scale, for the pellet pair with the largest preference difference, $IP_{A:C}$, and the product of the IPs from the other two pairs ($IP_{A:B} \times IP_{B:C}$). The within-session pairs of laser-on trials (yellow symbols) and laser-off trials (grey symbols) are indicated by a dark grey line connecting the scatterplot pairs. Histograms comparing laser-on and -off trials are shown in the background for each of the plots. The six sessions shown in (**A**) are marked with a red outline on the respective scatter points in (**B**) Data is provided in *Supplementary file 3* and *Figure 5—source data 1*. (**C**) The distances between within-session pairs (the lengths of the grey lines connecting the scatterplot pairs in (**B**) plotted as the cumulative empirical distribution function for each of the fiber-type conditions (blocked: yellow, patent: brown). For data see *Figure 5—source data 1*.

DOI: https://doi.org/10.7554/eLife.38963.011

The following source data is available for figure 5:

**Source data 1.** Text File Containing the Source Data for Figure 5 with the Same Layout as in *Supplementary file 3*.

DOI: https://doi.org/10.7554/eLife.38963.012

and the cumulative distribution of the lengths of these lines are shown in *Figure 5C*. To determine whether these distances were affected by inactivation, we ran a repeated measures ANOVA with the fiber-type as the single factor on the distances of the pairs of laser-on and –off trials and found no effect of Fiber: F(1,14) = 0.13, p=0.73.

Since we failed to find an effect of medial OFC inactivation on the economic choice task, we conducted two additional positive control experiments to be sure that the inactivation was working as expected. In one, we tested whether light would inhibit neurons in medial OFC in our subjects. After rats had completed behavioral testing, some rats (n = 3) were euthanized and slices were taken through medial OFC. We then performed whole cell current clamp recordings in neurons (n = 10) transfected with NpHR-YFP and assessed the effect of 593 nm light exposure (*Figure 6A*). Resting membrane potential was significantly hyperpolarized by 10 s light exposure (One-way ANOVA, F (1.06,9.49) = 26.42, p= 5.0e-4). The effect was maximal at the onset (peak) but maintained significant hyperpolarization in the steady state condition. In addition, repetitive firing activity elicited by a square pulse current injection of 400 pA was significantly inhibited by concurrent light exposure (Paired t-test, t(5) = 4.05, p=0.01). Following light exposure, direct current injection elicited fewer spikes per unit of current as evidenced by a rightward shift in the input-output curve (*Figure 6B*) (Two-way repeated measures ANOVA, F(11, 55)=4.3, p=1.0e-4).

With the remaining rats (n = 4), we tested whether light would impair performance on a progressive ratio task previously shown to be medial OFC-dependent in rats and mice (*Gourley et al., 2010*; *Münster and Hauber, 2017*) Rats were trained to lever press for reward for 2 days on a fixed ratio one schedule (FR1), after which they were tested on a progressive ratio schedule (PR5) across four sessions, using either the blocked or patent fiber (counterbalanced; *Figure 7A*) with the laser activated on all trials. A repeated measures ANOVA (rat and run were used as blocking factors) performed on the breaking point, corresponding to the number of pellets achieved, revealed a significant main effect of Fiber (F(1,10) = 9.97, p=0.01). This effect was robust with 7 of the eight blocked/patent pairs trending in the same direction (*Figure 7A* left). An ANOVA with the same design yet performed on the total lever presses also revealed a significant main effect of Fiber (F(1,10) = 7.85, p=0.02) (*Figure 7B* right). Importantly prolonged inhibition did not result in decreased motor activity; Figure (7B top) shows histograms of the inter-press intervals (IPIs) for the two conditions in which there appears to be no discernable difference. A two sample Kolmogorov-Smirnov test on the cumulative distributions of the IPIs (*Figure 7B* bottom) revealed no significant effect (D (1436,1045) = 0.028, p=0.74). Lastly, we checked whether medial OFC inhibition affected the animals' motivation to consume the food pellets. To do this, animals were given a bowl of 20 grams of food pellets in the conditioning box for 30 min to assess pellet consumption with and without inhibition. This test was repeated 4 times with blocked and patent cables (counterbalanced). A repeated measures ANOVA (with rat and run as blocking factors) showed no effect of inactivation on food consumption (F(1,10) = 0.20, p=0.66).

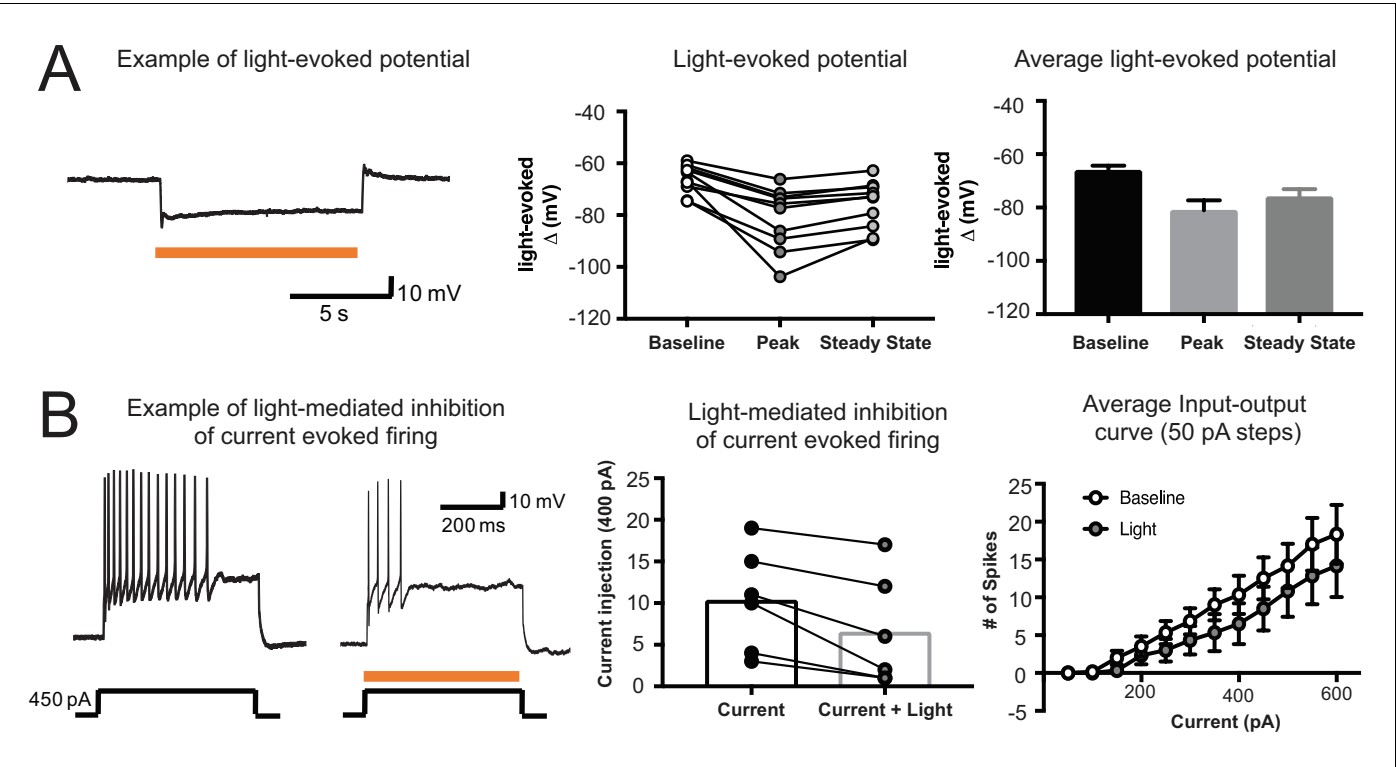

**Figure 6.** Confirmation of the Efficacy of Light to Inhibit Medial OFC Neurons Expressing Halorhodopsin in Slice Preparation. (**A**) Example trace showing hyperpolarization caused by 10 s light exposure in an OFC neuron (left). Summary graphs showing effect of 10 s light exposure on membrane potential for both the initial peak and the steady state conditions (right). (**B**) Example traces showing effect of current injection in an OFC neuron. When current injection is given with light exposure, firing frequency decreases (left). Summary graph showing spikes elicited by a 400 pA current injection alone or with light exposure (middle). Summary graph showing the effect of light exposure on the number of spikes elicited per unit of injected current.
DOI: https://doi.org/10.7554/eLife.38963.013

## Discussion

In the current study, we employed a rodent version of the economic choice task used to demonstrate neural correlates of economic value in primates (*Padoa-Schioppa and Assad, 2006*) to test whether medial OFC makes a critical contribution to economic choice behavior. We found that transient optogenetic inactivation of medial OFC on randomly selected trials had no effect. Inactivation in the same rats altered performance on a progressive ratio task known to be sensitive to medial OFC manipulations, and post-mortem testing confirmed that medial OFC neurons in these rats exhibited viral expression and light-sensitive reduction in spiking activity. Together these data show that the medial OFC is not necessary for economic choice in well-trained rats.

The current study builds on our prior report that optogenetic inactivation of lateral OFC also had no discernable effect on economic choice in this setting (*Gardner et al., 2017*). Inactivation of lateral OFC was motivated by influential reports that neural activity in lateral OFC correlates with value during economic choice (*McGinty et al., 2016*; *Padoa-Schioppa, 2009*; *Padoa-Schioppa and Assad, 2006*; *Padoa-Schioppa and Assad, 2008*; *Rich and Wallis, 2016*; *Tremblay and Schultz, 1999*; *Xie and Padoa-Schioppa, 2016*), and the associated proposal that lateral OFC was central to normal decision-making in this context (*Padoa-Schioppa, 2011*). Implicit in this proposal is that economic value is equivalent to the value revealed by devaluation underlying so-called model-based behavior. Such behavior typically depends on this part of OFC in both rats and primates (*Gallagher et al., 1999*; *Izquierdo et al., 2004*; *Machado and Bachevalier, 2007*; *Pickens et al., 2005*; *Pickens et al., 2003*; *Rudebeck et al., 2013*; *West et al., 2011*). Yet we found that we could inactivate lateral OFC and abolish changes in behavior due to devaluation without any effect on economic choice behavior.

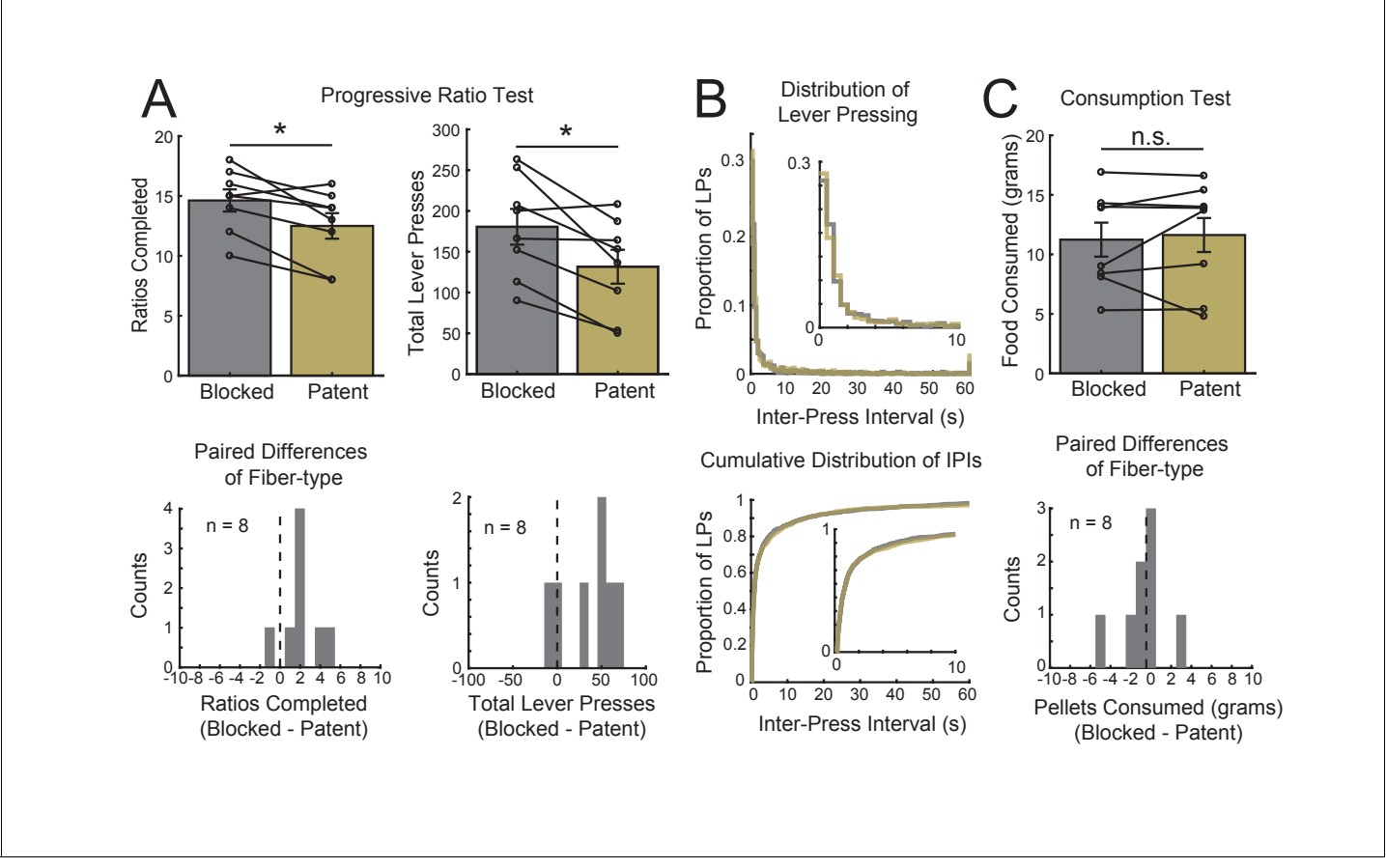

**Figure 7.** Confirmation of the Efficacy of medial OFC Inactivation on Behavior Using a Progressive Ratio Task. In order to determine whether optogenetic inactivation of medial OFC was sufficient to affect behavior, we inactivated medial OFC on the progressive ratio task which has been previously shown to be disrupted with medial OFC using pharmalogical inactivation. (**A**) Rats which were used in the choice task learned to press a lever for food delivery, FR1 schedule, and then were run for four subsequent days on a progressive ratio five schedule (see Materials and methods for details) with the fiber-types counterbalanced in a fully balanced design. The average break point (Y-axis, left, top) is shown for blocked (grey) and patent fibers (yellow) as well as the individual data points. Black lines show paired sessions (first two or last two sessions) for the two runs of each animal. A histogram of the paired differences is shown in the bottom left panel. The right two panels show the average lever presses (top) and histogram of the differences in lever presses from paired sessions (bottom). (**B**) Histograms of the inter-press interval for the blocked and patent conditions (bin size = 0.5 s, top), and the cumulative distribution of the inter-press intervals (bottom). Insets show only the 0 – 10 s bins of the plots. (**C**) To determine whether medial OFC inactivation reduced rats' motivation to consume pellets, we repeated the design used for the progressive ratio test, four consecutive days with fiber-types counterbalanced, while rats consumed pellets from a bowl for 30 min. Top panel shows the average pellets consumed over the 30 min (Y-axis) for each condition (blocked: grey, patent: yellow). Black lines show paired sessions (first two or last two sessions) for the two runs of each animal. A histogram of the paired differences is shown in the bottom panel. All bars represent mean ±SEM. *p<0.05 significance level.

DOI: https://doi.org/10.7554/eLife.38963.014

While this result did not support the lateral OFC as a critical, single bottleneck in which economic value is calculated, it left open whether other parts of the OFC might be performing this function. The OFC is now recognized to be a large area, potentially divisible into discrete subregions mediating discrete functions (*Heilbronner et al., 2016*; *Izquierdo, 2017*; *Murphy and Deutch, 2018*; *Rudebeck and Murray, 2011b*; *Walton et al., 2011*). Within this framework, the medial OFC is increasingly associated with representing value. While neural correlates of economic value have famously been reported in the spiking activity of single units in lateral OFC (*McGinty et al., 2016*; *Padoa-Schioppa, 2009*; *Padoa-Schioppa and Assad, 2006*; *Padoa-Schioppa and Assad, 2008*; *Rich and Wallis, 2016*; *Tremblay and Schultz, 1999*; *Xie and Padoa-Schioppa, 2016*), human imaging studies tend to find common currency correlates in the BOLD signal in medial OFC (*Levy and Glimcher, 2011*; *Plassmann et al., 2010*; *Plassmann et al., 2007*). Further, lesions to medial OFC in

monkeys disrupt value-guided decisions whereas lateral OFC lesions on the same behavior cause deficits in other more subtle functions, such as credit-assignment (*Noonan et al., 2010*). Other studies have shown that medial OFC is necessary for appropriate valuation during acquisition of cue-outcome pairings (*Baylis and Gaffan, 1991*), and also following selective satiation of one outcome during a choice task (*Rudebeck and Murray, 2011a*). Damage to medial OFC in humans is associated with similar deficits, as well as changes in economic decision-making (*Camille et al., 2011*; *Fellows and Farah, 2007*; *Noonan et al., 2017*). Indeed, the comparable part of OFC in rats, located along the base of the medial wall (*Heilbronner et al., 2016*; *Stalnaker et al., 2015*), has recently been shown to be important for using information about outcomes to guide behavior when those outcomes are hidden (*Bradfield et al., 2015*), risky (*Stopper and Floresco, 2014*), or temporally delayed, although there are conflicting results for this latter finding (*Mar et al., 2011*).

Yet despite the promising background data, we again failed to find any effect of inactivating the OFC on economic choice. Rats exhibited similar preferences among groups of flavored food pellets with or without medial OFC online, at least as indexed by indifference points, the steepness of the curves relating the two pellets, and the transitivity of the choices. This was true despite a reduction in the progressive ratio breakpoint in the same rats with inactivation, an effect that would traditionally be interpreted as an effect on valuation. These results show that medial OFC is also not the critical bottleneck for calculating a single common currency value to guide economic choice.

Instead, these and the prior data point strongly to the possibility that economic value and the resultant decision-making depend on parallel processing in multiple regions (*Hunt and Hayden, 2017*). Such parallel processing would be predicted if the value judgement underlying economic choice, at least as assayed in this task, reflected a variety of associative information (e.g. Pavlovian and instrumental, model-based and model-free, habitual and goal-directed). This is likely because the economic choice task used here, and those employed in other settings, generally do not try to limit or control the associative information at play. Tasks that employ such controls have clearly demonstrated that these different types of associative information do not generally converge into a single common brain region, but instead are mediated by somewhat non-overlapping brain circuits, each capable of mediating behavior somewhat independently, particularly in simple or poorly-controlled settings. Our results are more consistent with this sort of distributed representation of value.

Of course, this does not mean that the processing demonstrated in this and similar tasks in lateral and medial OFC is not important for economic choice behavior. Rather it simply means that a better controlled form of the task is necessary in order to reveal the critical contributions these areas (and others) each make to the very general function of economic choice. For example, it has been suggested that orbital areas are critical when choices are novel, rapidly changing, or involve elements that have never been experienced – such as the evocative tea-jelly and snail-porridge prepared by Behrens and colleagues (*Baylis and Gaffan, 1991*; *Rudebeck and Murray, 2011b*; *Fellows and Farah, 2007*; *Noonan et al., 2010*); *Chau et al., 2014*; *Barron et al., 2013*). Viewed in this light, the dissociation here (and also in our prior study) between economic choice and OFC-dependent behaviors, such as progressive ratio and Pavlovian devaluation for lOFC, occurs because these behaviors require a form of inference or estimation that is not necessary in a well-trained subject in the standard economic choice task like that used here.

Importantly, while this model of economic value takes us away from one in which all information is compressed into a single common currency value in a given brain area, this idea may return to be relevant in a more limited way once the precise type of information determining the role of these areas in the task can be isolated. Economic choices will depend on a particular region to the extent the choice emphasizes the processing dependent on that area. In this regard, the role of different subregions within OFC in economic choice behavior would be identical to their roles in other value-based behaviors.

## Materials and methods

**Key resources table**

| Reagent type (species) or resource | Designation | Source or reference | Identifiers | Additional information |
|---|---|---|---|---|

*Continued on next page*

*Continued*

| Reagent type (species) or resource | Designation | Source or reference | Identifiers | Additional information |
|---|---|---|---|---|
| Strain, strain background (*Rattus norvegicus*) | Long-Evans Rat | Charles River | RRID: RGD_2308852 | |
| Transfected construct (*Rattus norvegicus*) | AAV5/CamKIIa-eNpHR3.0-eYFP | UNC Vector Core | | |
| Chemical compound, drug | DAPI-Fluorescent - G | Electron Microscopy Services | Cat No. 17984–24 | |
| Chemical compound, drug | Triton X-100 | Sigma-Aldrich | Cat No. X100-500ML | |
| Software, algorithm | MATLAB | Mathworks | RRID: SCR_001622 | |
| Other | Doric dual optical commutators | Doric Lenses | Cat No. FRJ_1 × 2i_FC-2FC_0.22 | |
| Other | 200 micron diameter fiber optic patch cable | Thor Labs | M72L01 | |
| Other | Fiber optic cannulae | Thor Labs | Cat No. CFM12U-20 | |
| Other | ceramic zirconia ferrule bore 230 um | Precision Fiber Products | Cat No MM-FER2002S15-P | |
| Other | FC multimode connector | Precision Fiber Products | Cat No. MM-CON2004-2300-2-BLK | |
| Other | 543 nm DPSS Laser | Shanghai Lasers | Cat No. GL543T3-100 | |
| Other | Arduino Mega | Adafruit Industries | Cat No. 191 | |
| Other | 3.5' Resistive Touch Screen | Adafruit Industries | Cat No. 2050 | |
| Other | Raspberry Pi 3 B | Adafruit Industries | Cat No. 3055 | |

## Subjects

Six male Long-Evans rats (275 – 300 g, Charles River Laboratories), aged approximately 3 months at the start of the experiment, were trained and tested at the National Institute on Drug Abuse Intramural Research Program (Baltimore, MD) in accordance with the National Institute of Health guidelines determined by the Animal Care and Use Committee. All rats had ad libitum access to water during the experiment and were fed 16 – 20 grams of food per day, including rat chow and pellets consumed during the behavioral task. Rats were initially food restricted to 85% of their baseline weight to begin training. Behavior was performed during the light phase of the light/dark schedule.

## Apparatus

Rats were trained and tested in modified standard behavioral boxes (12' x 10' x 12', Coulbourn Instruments, Holliston, MA) that were controlled by a Raspberry Pi 3 (Raspberry Pi Foundation, Cambridge, UK) using custom-written code in Python (Python.org) which can be found at https://github.com/mphgardner/RatEconChoiceTask/blob/master/TS_Main.py (*Gardner, 2017*). Both custom-made equipment and Coulbourn components were used in the apparatus. Touchscreens (Adafruit Industries, New York, NY, 2.8' – initial training -and 3.5' – later training and testing) were housed in custom-made walls and were controlled by individual microcontrollers (Arduino Mega, Arduino, www.arduino.cc), which communicated with the Raspberry Pi 3 to display the current offers and

provide screen press feedback. Custom-designed nosepoke ports (1.5' H X 1.25' W X 1.5' D) with infrared photodetectors to determine whether a poke had occurred were fixed to the floor of the box about one inch from the wall. All cad files for custom parts are available at https://github.com/mphgardner/RatEconChoiceTask (*Gardner, 2017*; copy archived at https://github.com/elifesciences-publications/RatEconChoiceTask). The primary configuration of the box had touchscreens and accompanying wall mounts oriented at 30° from the plane of the left side wall to facilitate better viewing of the screen while the rats were nosepoking at the central port. A tall recessed food magazine (Med-Associates, Fairfax, VT) was placed on the center of the right wall opposite to the nosepoke and touchscreens. Pellets from two separate externally mounted feeders were dispensed into the food magazine. The speaker used for playing the white noise cue (75 dB) to indicate the beginning of a trial was placed externally to the conditioning chamber. During the optogenetic inhibition phase of the experiment, solid state lasers (532 nm; Laser Century, Shanghai China) were controlled in analog mode (8 bit depth) by a microcontroller (Arduino Uno, Arduino, www.arduino.cc)

## Choice task

Each trial started with a white noise cue, which indicated that the rat could nosepoke at the central port. After a 1 s nosepoke at the port, the current offers were displayed on the two screens situated on either side of the nosepoke. After another 1 s period, during which the rats were required to remain in the nosepoke, the white noise ended indicating that a choice could be made by touching either of the screens to receive the offer-type and pellet number displayed. Immediately following the choice, the pellets were delivered into the food magazine on the opposite side of the chamber. Rats then waited 6 – 16 s before the next trial started which depended on a random component as well as the number of pellets delivered on the prior trial. This was determined empirically such that rats were not waiting for longer periods of time for the next trial to start following trials in which only 1 or 2 pellets were delivered. Failure to hold the nosepoke for the first second restarted the 1 s timer and failure to hold the nosepoke once the screens were displayed resulted in the termination of the trial. Rats performed ~150 – 350 trials per session. Data is available at https://github.com/mphgardner/Econ-Choice-Task (*Gardner, 2018*; copy archived at https://github.com/elifesciences-publications/Econ-Choice-Task).

All rats received the same menu of pellet offers arranged in the following average preference order (highly palatable banana flavored pellets, Test-Diet 5-TUL (1813985); bacon flavored pellets containing lactose and 1.4% NaCl, Bio-Serv, custom formulation (F07382); grain flavored pellets, Test-Diet 5-TUM (1811143); grape flavored pellets with 50% sucrose and 50% cellulose, Test-Diet, custom formulation (1817455 – 371); chocolate flavored pellets with 25% sucrose and 75% cellulose, Test-Diet, custom formulation (1817259 – 371); and 100% cellulose pellets, Test-Diet 5-TUW (1811557). Visual cues predicting the different offer-types consisted of different shapes, indicating the type of pellet available, and different numbers of segmentations of the symbol, indicating the number of pellets available, see (*Gardner et al., 2017*) for visual cues used. Each rat received unique cue-pellet pairings that remained constant throughout testing.

## Shaping and Pre-Surgical training

Initial training on the task lasted 3 – 4 months before rats experienced any of the tested pairs of pellets and progressed through several stages that introduced different aspects of the task. Before starting, rats were food restricted to ~85% of their body weight, then they were first trained to touch a single illuminated touchscreen to receive unflavored sucrose pellets, after which they began training to discriminate two visual cues which either resulted in an unflavored sucrose pellet or nothing (the images used were not used for any subsequent aspect of the task). After rats showed discriminative behavior to the two visual cues, a central nosepoke was introduced to the box and rats were progressively trained to hold in the port for 2 s (1 s with no cues on and one second with visual cues displayed) when the white noise cue was turned on. Upon acquisition of the nosepoke, rats were introduced to the full task. To learn each of the cue-pellet associations, rats were trained for several days on each of the five flavored pellets versus a non-preferred cellulose pellet. After rats showed stable preferences for each of the pellets versus cellulose, they were exposed to other pellet-pairs. In each session, rats were given 11 possible offers including the 1:0 and 0:1 offers. The other 9 offers ranged either from 1:6 to 6:1 or 1:4 to 8:1 (X:Y, Y being the presumed preferred pellet-type) from

the offer set [1:8, 1:6, 1:4, 1:3, 1:2, 1:1, 2:1, 3:1, 4:1, 6:1, 8:1] depending on the presumed pair preference.

## Surgery

Surgical procedures followed guidelines for aseptic technique. Rats received AAV-CaMKIIa-eNpHR3.0-eYFP (Gene Therapy Center at University of North Carolina at Chapel Hill) bilaterally into the medial OFC under stereotaxic guidance at AP 4.7 mm, ML ±0.6 mm, and DV −3.6 mm from the brain surface. A total 1 µl of virus (titer ~$10^{12}$) per hemisphere was delivered at the rate of ~0.1 µl/min by infusion pump (*Takahashi et al., 2013*). Immediately following viral infusions, optic fibers (200 µm in core diameter; Thorlab, Newton, NJ) were implanted bilaterally at A/P: 4.7 mm, M/L: ±0.6 mm, and D/V:- −3.4 mm (from dura) at an angle of 12 degrees in the M/L plane (*Bradfield et al., 2015*). Cephalexin (15 mg/kg p.o.) was administered daily for 10 days post-operatively to prevent infection.

## Post-Surgical testing

Following a 2 – 3 week recovery from surgery, rats were food restricted and then retrained on the full task and acclimated to performing full sessions reliably with two fiber optic patch cables attached to an optic commutator (Doric Lenses, Quebec Canada). During this period of training, which required ~5 weeks, rats were given sessions in which the flavored pellets were paired with cellulose, to avoid additional over-training. Cables were constructed with blocking covers to reduce leakage of light into the box. However, it is impossible to completely eliminate light leakage. To control for effects of such light leakage during laser-on trials, 'dummy' patch cables were employed during retraining and testing. The 'dummy' patch cables were identical to the patent patch cables except that the optical fiber was blocked at the end of the cable and permitted no light transmittance into the brain. The 'dummy' patch cables were constructed identically to the patent cables with one exception; the optical fiber was terminated at the ferrule, or ~1 cm, from the animal-side terminal of the patch cable. A solid metal wire was inserted into the ferrule and epoxied into place in order to block. All 'dummy' cables were tested after construction as well as on a periodic basis using a Fiber Optic Power Meter (ThorLabs). After rats were familiarized with the 'dummy' patch cables and the laser being turned on for 50% of trials, inactivation testing was begun (~8 weeks after viral infusion). Testing proceeded in 9 day blocks in which rats were given each possible pair of three pellet-types for three days. The first day on an offer pair the blocked, or 'dummy', patch cables were used. The following two days on the pellet-pair rats had counterbalanced 'dummy' cable or patent cable providing a within-subjects control for light in the brain. Stepping through the pairs of three pellet types allowed for the collection of transitivity data across three different pellets which we refer to as a triplet. In a few cases in which rats removed the patch cable during a session, the three day period was thrown out and an additional three days were provided for the particular offer-pair at the end of the 9 days. In all sessions the laser was turned on for 50% of trials during the cue period of the trial. This experimental design was repeated three times for each rat. The specific pellets used for each of the three repetitions are as follows (Run 1: Bacon, Grain, Chocolate; Run 2: Banana, Grain, and Grape; Run 3: Banana, Bacon, Chocolate; see above for detailed description of the pellets). Lasers (532 nm, 16 – 18 mW; Laser Century, Shanghai China) were controlled by a microcontroller (Arduino Uno, Arduino) and were turned on concurrently with the white noise cue to indicate the availability to begin a trial. Lasers were turned off at the time of decision using a linear ramp over 300 ms to avoid the possibility of rebound excitation. To minimize the duration of the laser, the white noise and laser were on for 5 s before a timeout period occurred. Rats also had a maximum of 5 s to make a choice once nosepoke hold was fulfilled. Sessions lasted 2 – 2.5 hr.

## Progressive ratio task

Four of the six rats (two rats were immediately used to confirm expression and functionality of the halorhodopsin virus) used in the touchscreen task were subsequently tested on the progressive ratio task. A lever was inserted into the back wall of the operant chamber used for the touchscreen task, and rats were trained for two days on a fixed ratio 1 (FR1) schedule for 30 min such that each press resulted in the delivery of a 45 mg highly appetitive pellet (5-TUL, TestDiet). All rats displayed vigorous lever pressing by the end of the second day of FR1 schedule training. Rats were then tested on

a progressive ratio task for four consecutive days with either blocked or patent cables, counterbalanced across animals and days. In these sessions, the laser was turned on, and the number of lever presses required to receive a pellet increased every three trials according to the schedule (1,5,10,15,20,...). The session was terminated either after 30 min or when the animal ceased lever pressing for 5 min. The number of trials reached was used as the experimental measure for the study (*Münster and Hauber, 2017*; *Schweimer and Hauber, 2005*). Data is available at *https://github. com/mphgardner/Econ-Choice-Task* (*Gardner, 2018*).

For the consumption test, rats were given 20 grams of the same pellet-type used for the progressive ratio task in a bowl placed in the operant box. Food consumption was measured after 30 min. Rats were tested for 4 days with one day off between each test in order to maintain a consistent body weight.

## Histology

After completion of the experiment, rats were perfused with phosphate buffer saline followed by 4% PFA. The brains were then immersed in 30% sucrose for at least 24 hr and frozen. The brains were sliced at 40 μm and stained with DAPI (through Vectashield-DAPI, Vector Lab, Burlingame, CA). The location of the fiber tip and NpHR-eYFP or eYFP expression was verified using an Olympus confocal microscope.

## Ex vivo brain slice electrophysiology

Rats were deeply anesthetized with isoflurane and transcardially perfused with an ice-cold solution containing (in mM) 93 NMDG, 93 HCl, 2.5 KCl, 1.2 $NaH_2PO_4$, 30 $NaHCO_3$, 20 HEPES, 25 Glucose, 5 Na-ascorbate, 2 Thiourea, 3 Na-pyruvate, 10 $MgSO_4$, 0.5 $CaCl_2$. Brains were rapidly removed, and coronal OFC slices (250 μm) were made using a vibratome (Leica VT-1000S). Slices were then incubated in a holding solution containing (mM) 92 NaCl, 2.5 KCl, 1.2 $NaH_2PO_4$, 30 $NaHCO_3$, 20 HEPES, 25 Glucose, 5 Na-ascorbate, 2 Thiourea, 3 Na-pyruvate, 2 $MgSO_4$, 0.5 $CaCl_2$(32 – 34°C) for 15 – 30 min. Following this, the holding chamber was kept at room temperature for the duration of the experiment. Slices were transferred to a recording chamber and superfused (2 – 3 mL/min) with artificial cerebrospinal fluid containing (in mM) 126 NaCl, 2.5 KCl, 1.2 $MgCl_2$, 2.4 $CaCl_2$, 1.2 $NaH_2PO_4$, 21.4 $NaHCO_3$, 11.1 glucose maintained at 32 – 34°C. All solutions were continually oxygenated (95% oxygen, 5% carbon dioxide). Glass pipettes (tip resistance 2 – 4 MΩ) were used for whole cell recordings and were filled with an intracellular solution containing (mM) 115 K-gluconate, 20 KCl, 1.5 $MgCl_2$,. 025 EGTA, 10 HEPES, 2 Mg-ATP, 0.2 Na-GTP, 10 $Na_2$-phosphocreatine (pH 7.2 – 7.3, ~290 mOsm/kg).

Virus-infected (eYFP+) cells were identified using scanning disk confocal microscopy (Olympus FV1000), and differential interference contrast optics were used to patch neurons. Whole cell current clamp recordings were performed in visually identified neurons in the orbitofrontal cortex. For NpHR experiments, a 593 nM laser (OEM laser systems; maximum output 150 mW) attached to fiber optic cable was used to deliver light to the slice. Light intensity of 8 – 12 mW was used to stimulate NpHR in slice recordings. For experiments shown in *Figure 6A,* a 10 s light pulse was delivered to the slice. For experiments shown in *Figure 6B*, current pulses were injected (500 ms square pulse, 50 pA-600 pA). Half of the time, a 500 ms light pulse was given during current injection. Images were acquired using Olympus Fluoview software (version 3.0).

Electrophysiology data were acquired using an Axopatch 200 B amplifier (Molecular Devices, San Jose, CA) in either voltage clamp or current clamp mode. Axograph X software (Axograph Scientific) were used to record and collect the data, which were filtered at 10 kHz and digitized at 4 – 20 kHz. Series resistance (Rs) was monitored with injection of hyperpolarizing current (−20 pA, 500 ms) and data were excluded if Rs changed >20% during data acquisition. Recordings were discarded if series resistance or input resistance changed >10% throughout the course of the recording.

## Quantification and statistical analysis

Raw data was collected using custom written code in Python for both the economic choice task and for the progressive ratio task. All further analysis was performed using MATLAB. As described previously (*Padoa-Schioppa and Assad, 2006*), in order to estimate a scalar relative value of two goods from a limited subset of all possible offers, an assumption must be made about the function relating

the two goods in offer space. Here we assume a linear indifference curve (within a reasonable set of offer space) which entails that the ratio of the number of each good offered leading to indifferent behavior remains constant as the number of goods offered increases. In order to estimate the relative value of two goods from the choice behavior we performed a probit regression for each session (*Padoa-Schioppa and Assad, 2008*), which uses the cumulative distribution function of the normal distribution to predict the choice behavior given the log ratio of the offers. This provides estimated parameters $\mu$ and $\sigma$ of the fitted normal distribution, which were used as estimates for the log of the indifference point (*IP*) - the estimated relative value - and inverse slope parameter respectively. Sessions with relative pellet values outside of the offer range tested were not included in the analysis. This was done by excluding sessions (n = 4) with estimated indifference points $IP = exp(\hat{\mu})$ greater than a 6:1 ratio (non-preferred:preferred pellet).

### Indifference Point and inverse slope

The average choice behavior across sessions (*Figure 2B*) was computed by subtracting the log indifference point from the log of the offer ratios for each session. The relative offer ratios were then binned into the intervals shown in *Figure 2B*. A probit regression was then performed on the aligned and averaged choice data for visual comparison. To test the effect of inactivation of medial OFC on the indifference point and the inverse slope, a fully balanced design was implemented with fiber-type (blocked/patent) and laser (on/off) as the factors in a two-way ANOVA. The number of repetitions of the design (n = 50) was used to match our previous paper of inactivation of lateral OFC (*Gardner et al., 2017*). As noted above, some sessions (n = 4) were dropped due to IPs exceeding a 6:1 ratio. Repetitions of the design were also used as a blocking factor due to the wide range of subjective indifference points for each rat and pellet-pair. This block factor had a main effect of p<0.01 in all cases. This test was repeated on a subset of the sessions in which rats showed a significant preference for one of the pellets. In order to determine the threshold for indifference points sufficiently far from the 1:1, trials from sessions with the blocked fiber were randomly split into two separate groups. A bootstrap on the difference between the log of the indifference point for each group was used to determine a significance threshold ($\alpha = 0.05$) for indifference point shifts.

### Transitive behavior

Assuming linear indifference curves, transitivity is defined as the log of the relative values of three goods A, B and C (in that preferred order) having the following relation, $logIP_{A:C} \approx logIP_{B:C} + logIP_{A:B}$ (*Padoa-Schioppa and Assad, 2008*). In order to compare the transitive behavior during the inactivation experiment, the transitivity measure was compared in a two-way ANOVA with factors: laser (on/off) and cable-type (blocked/patent). The distance between paired points on the transitivity plot for laser-on and -off trials from the same sessions was also computed and compared using a two-way ANOVA with fiber-type and virus used as factors

### Progressive ratio task

A within-subjects design was implemented using four of the animals used in the economic choice task. The sample size (n = 8) was constrained by the number of rats used in the economic choice experiment and was ultimately chosen based on prior work showing a positive effect of medial OFC lesions on the PR task (*Münster and Hauber, 2017*). The number of ratios completed and the number of lever presses for each session were used as the measure of interest. A repeated measures ANOVA was used to compare sessions with the blocked and patent fibers with rat and run used as blocking factors. The same analysis was run for the consumption test.

## Acknowledgments

This work was supported by the Intramural Research Program at NIDA (GS). The authors thank Hannah Batchelor and Marlian Montesinos for their help with histology; The authors would like to acknowledge the Ex Vivo Electrophysiology Core facility at NIDA IRP; Andrew Wikenheiser, Brian Sadacca, Melissa Sharpe, and Jingfeng Zhou for their helpful feedback; and Dr. Karl Deisseroth and the Gene Therapy Center at the University of North Carolina at Chapel Hill for providing viral reagents. The opinions expressed in this article are the authors' own and do not reflect the view of the NIH/DHHS.

## Additional information

### Funding

| Funder | Grant reference number | Author |
|---|---|---|
| National Institute on Drug Abuse | Intramural Research Program | Geoffrey Schoenbaum |

The funders had no role in study design, data collection and interpretation, or the decision to submit the work for publication.

### Author contributions

Matthew PH Gardner, Conceptualization, Resources, Formal analysis, Supervision, Funding acquisition, Methodology, Writing—original draft, Project administration, Writing—review and editing; Jessica C Conroy, Conceptualization, Data curation, Software, Formal analysis, Validation, Investigation, Visualization, Methodology, Writing—original draft, Project administration, Writing—review and editing; Clay V Styer, Data curation, Software Development, Investigation, Methodology; Timothy Huynh, Software, Investigation; Leslie R Whitaker, Investigation, Methodology; Geoffrey Schoenbaum, Resources, Data curation, Formal analysis, Investigation, Visualization, Methodology, Writing—original draft, Project administration, Writing—review and editing

### Author ORCIDs

Matthew PH Gardner (iD) http://orcid.org/0000-0002-9146-5043
Geoffrey Schoenbaum (iD) http://orcid.org/0000-0001-8180-0701

### Ethics

Animal experimentation: Animals were trained and handled at the National Institute on Drug Abuse Intramural Research Program (Baltimore, MD) in accordance with the National Institute of Health guidelines determined by the Animal Care and Use Committee. protocol 15-CNRB-108 and following the guidelines provided by the Guide for the Care and Use of Laboratory Animals of the National Institutes of Health. All surgical procedures were performed under isoflurane anesthesia and all efforts were taken to minimize suffering.

### Decision letter and Author response

Decision letter https://doi.org/10.7554/eLife.38963.020
Author response https://doi.org/10.7554/eLife.38963.021

## Additional files

### Supplementary files

• Supplementary file 1. Three-way ANOVA Results Comparing Medial OFC Inactivation with Behavioral Measures from a Virus Control Group.
DOI: https://doi.org/10.7554/eLife.38963.015

• Supplementary file 2. Results of Two-Way ANOVA with Factors Laser and Fiber Performed on a Subset of Trials Taken from the Beginning of each Session.
DOI: https://doi.org/10.7554/eLife.38963.016

• Supplementary file 3. Indifference Points for Sets of Transitivity Measures across Three Pellet-Types.
DOI: https://doi.org/10.7554/eLife.38963.017

• Transparent reporting form
DOI: https://doi.org/10.7554/eLife.38963.018

### Data availability

Data is available in Github (https://github.com/mphgardner/Econ-Choice-Task) as well as in the source data file for Figure 5.

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
