## [Decision Letter]

[Editors’ note: this article was originally rejected after discussions between the reviewers, but the authors were invited to resubmit after an appeal after it was made clear that the key issue was based on a misunderstanding of the course of training and was therefore unfounded]

Thank you for submitting your work entitled "Medial orbitofrontal inactivation does not affect economic choice" for consideration by *eLife*. Your article has been reviewed by three peer reviewers, and the evaluation has been overseen by a Reviewing Editor and a Senior Editor. The following individuals involved in review of your submission have agreed to reveal their identity: Kevin Miller (Reviewer #2).

Our decision has been reached after consultation between the reviewers. Based on these discussions and the individual reviews below, we regret to inform you that your work will not be considered further for publication in eLife. The overall conclusion is that this work is potentially of interest and could be worthy of publication in eLife in the future providing the concerns can be fully addressed.

The key issue is that the main result of the paper is a negative result and that it is not clear that you have sufficient evidence that the inactivation was effective. This is particularly important because the expression time, as far as we understand, was short for the test protocol, but long for the positive controls. This point is made most clearly in point 1 from Reviewer #3.

Reviewer #1:

The idea that the orbitofrontal cortex (OFC) is central to computing and then comparing economic values has been one of the leading ideas about the function of this brain region. While a wealth of correlative functional imaging and neurophysiology abound, causal evidence for this hypothesis has, however, not been so clear-cut. For instance, deficits in the ability to update the value of outcomes in reinforcement devaluation task after OFC lesions have been taken as evidence for the OFC in economic choice, but this task requires a learning component muddying whether this paradigm is really assessing economic choice.

To get around this issue, Gardiner and colleagues developed a rodent version of the task that has classically been used by Padoa-Schioppa to demonstrate the role of OFC in economic choice behaviors. In single sessions, rats are offered choices between two stimuli associated with different amounts of flavored food pellets. Each day the rat's subjective equivalence curves are determined by systematically varying the number of each pellet available on each trial. Previously this group showed that inhibition of lateral OFC had no effect on performance of this task: rat's subjective equivalence curves did not change after inhibition. This is directly counter to the idea that OFC is required for economic choices. Here they tested whether another part of OFC, medial OFC might be the area required for economic choice. Prior work suggested that this area was required for comparing values, so maybe this is the critical computation/area for that their task is testing. Using precisely controlled optogenetic stimulation the authors show that inhibition of medial OFC does not alter economic choice behavior. Instead, inhibition is associated with a decrease in the number of competed trials in a progressive ratio test known to be sensitive to OFC inhibition.

This study is well motivated by the current state of the field, the experiments cleverly designed and the results are clear. It follows on from their previous work on OFC contributions to economic choice and is a strong compliment to that work. I only have a small comment for them about the interpretation of their effects as the paper exhaustively analyzes the data and there isn't much need for more in my opinion.

The null effect of medial OFC inhibition on economic choice task performance is very clear-cut, but this then raises the question of what makes medial OFC necessary for progressive ratio performance, which is a cost/benefit economic type decision. The authors touch on this a little in the Discussion, but I'm really interested to know why the authors think that medial OFC can be involved in some economic-like choices but not others? This question is motivated by the fact that a number of studies have shown that the medial OFC is required for economic type choices but only when the items compared have not been compared before (Baylis and Gaffan, 1991; Rudebeck and Murray, 2011; Fellows and Farah, 2007), are being affected by a bad other options that are constantly changing (Noonan et al., 2010; Chau et al., Nature Neurosci., 2014), or choices are between food combinations that are novel, like choosing between tea jelly and snail porridge (Barron et al., Nature Neuroscience, 2014).

So, there are two points here: 1) In the current design the type of pellets being considered on each trial are known to the animal, but at the start of the session the comparison is somewhat novel. If only the first few trials are looked at as opposed to the whole session, do the authors see any difference between the optogenetic inhibition trials and control trials? There may well not be enough power to properly run this analysis because the laser delivery is random from trial-to-trial, but I'm interested to know if there might be something there. It is also the case that as the first day of the three days that the animals get a specific pair of rewards is a control day the novelty of the comparison may have already passed, rendering the above futile. 2) Given the above literature on medial OFC in the comparisons of novel/changing options, I think it is worth noting how this factors into their multiple systems model of decision making that the authors highlight in the Discussion. Ultimately a rat may not need a medial OFC for this type of comparison because the available options are well learned and the decisions relatively automatic.

*Reviewer #2:*

This is a well-executed and careful study investigating the brain regions that are (or aren't) necessary for economic decision-making. In previous work, the authors developed a rat version of the iconic Padoa-Schioppa economic choice task, and demonstrated that inactivations of lateral OFC do not affect performance on this task. In this paper, they extend their approach to medial OFC, finding that this region also is not necessary for normal behavior on the task. The authors show that this is true across a variety of behavioral measures from a relatively rich task, and do a good job of validating their inactivation using manipulation checks. Extending their findings to mOFC represents an important contribution to the literature, and casts further doubt on the influential idea that OFC plays a key role in value-based decision-making. It is especially important given lesion data from monkeys and humans (Noonan, et al., 2010; Noonan et al., 2017) which has suggested a role specifically for mOFC, but not lOFC, in value-guided choice per se. Together with the literature, this study motivates the search for more nuanced and general theories about the role of the OFC and its subregions in value-guided behavior.

I have two relatively minor suggestions, which I believe would be straightforward to implement and would improve the paper:

1) In this study, the authors do not include a control group of animals injected with a YFP-only virus, as all comparisons necessary to support their claims are within-subject, and the control animals from their previous study showed no effect. This seems very reasonable. But why not present statistics directly comparing the rats from this study to the control rats from the previous study? If the task details are the same (my understanding is that they are) this should be a fair comparison, and would strengthen the claims made here.

2) The main findings of this paper are supported by a series of statistical tests which fail to reject their null hypothesis that mOFC inactivation does not affect behavior. The strength of these findings is therefore sensitive to the size of the dataset that was collected (failing to reject the null with thirty rats is stronger than failing to reject it with three). It seems to me that an important statistic to report in this case is the largest effect size that *can* be rejected based on the authors' dataset. For example, by eyeballing Figure 2B, it seems clear that the data rule out the hypothesis that inactivating mPFC drives the indifference point all the way back to 1:1. But do they rule out the hypothesis that it is driven halfway back? Five percent of the way back? Clearly there is some range of small (but nonzero) effect sizes that cannot be rejected by the authors' dataset – it would be helpful to compute and report this range. An alternative approach addressing the same concern would be to perform a Bayesian model comparison considering both the null and an explicit alternative hypothesis (see e.g. Gallistel, 2009, The Importance of Proving the Null; Wagenmakers et al., 2010, Bayesian Hypothesis Testing for Psychologists), and reporting the strength of evidence in favor of the null.

*Reviewer #3:*

In this study the authors set out to determine whether the medial orbitofrontal cortex (mOFC) is responsible for computing and comparing the subjective value of choice options thus enabling economic decisions. Rats learned to choose between pellets that varied in both flavor and number to determine the indifference point (IP) for which the choice probability and, in turn, the subjective value between pellet options is equal. The authors demonstrate that bilateral optogenetic inhibition of mOFC during stimulus presentation has no effect on the IP or other choice metrics such as preference stability. The authors confirmed the efficacy of the inhibition protocol via slice electrophysiology and additionally showed that mOFC inhibition disrupted a progressive-ratio task. In conclusion, the authors demonstrate evidence that mOFC is not necessary for subjective-value comparisons in well-trained rats. This study contributes an important piece of evidence towards understanding OFC's role in economic decisions. The task allows for probing subjective-value comparisons and the authors convincingly argue that mOFC inhibition does not impact various aspects of the behavior. I have a few specific questions to be addressed in revision:

1) The main claim is a negative result and hence it is critical to establish that their inactivation worked. The authors state that behavior testing began 2-3 weeks after injection and implantation. This seems rather short to me for AAV-CaMKIIa-eNpHR expression. Do they have a demonstration of full efficacy of inhibition after this expression time? Ideally electrophysiological evidence should be provided that they managed to inactivate mOFC. Both 'positive controls' (slice and progressive-ratio test) were conducted weeks after the value task, i.e. with substantially longer expression times, hence they are expected to have larger effect size. They should first perform the progressive-ratio test before the economic decision.

2) In the progressive ratio task, the effect size of mOFC inhibition seems rather small (~2 trials), although consistent (Figure 7). What are comparable effect sizes reported in other manipulations? In reference to the PR task (Figure 7), where does the n=8 and F(1,10) come from?

3) It is not fully clear how disrupted economic choice would look like for specific behavioral metrics. What are the hypotheses regarding a disruption or changes in this task? For example, a change in IP would reflect a change in subjective value between choice options. Would inhibition of a value-computing area during parallel stimulus presentation expected to shift IP? Flatten IP function? Etc.

4) Has there been a demonstration that any manipulation can shift the IP?

5) Please clarify the pair contingencies in the Materials and methods section. Specifically, for 'transitivity' tests, how were the pellet flavor pairs chosen? What are the respective IPs for the chosen triplets for the six rats.

6) I did not fully understand the construction of the blocked-light patch cable.

---

## [Author Response]

[Editors’ note: the author responses to the first round of peer review follow.]

We have now clarified the training schedule to make clear that the critical testing occurred across a number of weeks, starting ~8 weeks after viral infusion and not 2-3 weeks as implied in the prior text. We have also split our analyses into thirds, to show no effect at any time point in this prolonged training, and we have conducted power and other analyses to give some idea of the strength of our result.

Reviewer #1 (General assessment and major comments (Required)): […] This study is well motivated by the current state of the field, the experiments cleverly designed and the results are clear. It follows on from their previous work on OFC contributions to economic choice and is a strong compliment to that work. I only have a small comment for them about the interpretation of their effects as the paper exhaustively analyzes the data and there isn't much need for more in my opinion. The null effect of medial OFC inhibition on economic choice task performance is very clear-cut, but this then raises the question of what makes medial OFC necessary for progressive ratio performance, which is a cost/benefit economic type decision. The authors touch on this a little in the Discussion, but I'm really interested to know why the authors think that medial OFC can be involved in some economic-like choices but not others? This question is motivated by the fact that a number of studies have shown that the medial OFC is required for economic type choices but only when the items compared have not been compared before (Baylis and Gaffan, 1991; Rudebeck and Murray, 2011; Fellows and Farah, 2007), are being affected by a bad other options that are constantly changing (Noonan et al., 2010; Chau et al., Nature Neurosci., 2014), or choices are between food combinations that are novel, like choosing between tea jelly and snail porridge (Barron et al., Nature Neuroscience, 2014).

This is a really interesting question. As noted, we simply used progressive ratio as a confirmation, since mOFC in rats had been shown to be necessary for normal break points and this task does seem related to value and choice behavior. The idea that it might only be involved in comparisons between novel items is a point well-taken. Though if this were the case, it seems to us that mOFC would also not be necessary for progressive ratio, since this does not involve comparing new items. Unless the progressive omission of rewards is considered a novel option. It might be seen in this light. We now make this point in the Discussion.

So, there are two points here: 1) In the current design the type of pellets being considered on each trial are known to the animal, but at the start of the session the comparison is somewhat novel. If only the first few trials are looked at as opposed to the whole session, do the authors see any difference between the optogenetic inhibition trials and control trials? There may well not be enough power to properly run this analysis because the laser delivery is random from trial-to-trial, but I'm interested to know if there might be something there. It is also the case that as the first day of the three days that the animals get a specific pair of rewards is a control day the novelty of the comparison may have already passed, rendering the above futile. 2) Given the above literature on medial OFC in the comparisons of novel/changing options, I think it is worth noting how this factors into their multiple systems model of decision making that the authors highlight in the Discussion. Ultimately a rat may not need a medial OFC for this type of comparison because the available options are well learned and the decisions relatively automatic.

This is a great idea. Though the rats are, of necessity, trained on the comparisons before the test, they may go a few weeks between experiencing a specific pair, so as the reviewer points out, once the animal re-experiences a pair, the decisions are ‘somewhat’ novel in that they have not chosen between the two particular pellets for a considerable time. So it is an interesting question of whether the differences might be there early. However actually looking at this is a bit difficult since the data is a series of Bernoulli trials spread across 9 offers – and so necessarily many trials are needed to get a reasonable estimate of the IP. We did perform this analysis using an n = 4 for each offer – on average this is equivalent to the initial 29.9% of the session. We again found no effect of inactivation, and these results are now reported in the paper.

Of course this is not a fair test of the general question. So we have adjusted our Discussion to better incorporate this idea, in the same place we address the same question about the progressive ratio impairment.

Reviewer #2:[…] I have two relatively minor suggestions, which I believe would be straightforward to implement and would improve the paper:1) In this study, the authors do not include a control group of animals injected with a YFP-only virus, as all comparisons necessary to support their claims are within-subject, and the control animals from their previous study showed no effect. This seems very reasonable. But why not present statistics directly comparing the rats from this study to the control rats from the previous study? If the task details are the same (my understanding is that they are) this should be a fair comparison, and would strengthen the claims made here.

This is a great idea, so we have conducted this analysis and now include it in the revised manuscript. It does not change our conclusions, since there were no effects when we made this comparison. That said, I think it certainly adds strength to the argument when comparing to animals injected with a control virus. In order to be clear that the study was not designed with this as the plan, we have retained the original analysis and we now note this comparison to bolster the main result, with a supplementary figure.

2) The main findings of this paper are supported by a series of statistical tests which fail to reject their null hypothesis that mOFC inactivation does not affect behavior. The strength of these findings is therefore sensitive to the size of the dataset that was collected (failing to reject the null with thirty rats is stronger than failing to reject it with three). It seems to me that an important statistic to report in this case is the largest effect size that can be rejected based on the authors' dataset. For example, by eyeballing Figure 2B, it seems clear that the data rule out the hypothesis that inactivating mPFC drives the indifference point all the way back to 1:1. But do they rule out the hypothesis that it is driven halfway back? Five percent of the way back? Clearly there is some range of small (but nonzero) effect sizes that cannot be rejected by the authors' dataset – it would be helpful to compute and report this range. An alternative approach addressing the same concern would be to perform a Bayesian model comparison considering both the null and an explicit alternative hypothesis (see e.g. Gallistel, 2009, The Importance of Proving the Null; Wagenmakers et al., 2010, Bayesian Hypothesis Testing for Psychologists), and reporting the strength of evidence in favor of the null.

This makes a lot of sense and we have included some additional information for our tests. Most importantly, we’ve reported confidence intervals for the critical comparison of the laser-on and laser-off conditions for the patent cable – basically the effect size for which we would reject the null hypothesis. We also took reviewer #2’s suggestion of putting this in terms of a percent change of the preference.

Reviewer #3:[…] I have a few specific questions to be addressed in revision:1) The main claim is a negative result and hence it is critical to establish that their inactivation worked. The authors state that behavior testing began 2-3 weeks after injection and implantation. This seems rather short to me for AAV-CaMKIIa-eNpHR expression. Do they have a demonstration of full efficacy of inhibition after this expression time? Ideally electrophysiological evidence should be provided that they managed to inactivate mOFC. Both 'positive controls' (slice and progressive-ratio test) were conducted weeks after the value task, i.e. with substantially longer expression times, hence they are expected to have larger effect size. They should first perform the progressive-ratio test before the economic decision.

We appreciate the point the reviewer makes here. The statement that testing began 2-3 weeks after surgery was only meant to indicate the post-op recovery period. This is fairly standard for our lab. It was not meant to indicate the period between AAV infusion and actual inhibition of the OFC neurons. This period was substantially longer, not even beginning until 7-8 weeks following surgery. The intervening 4-5 weeks was spent food restricting the rats again, re-exposing them to each food pellet against cellulose, and acclimating them to the task with the patch-cables attached. Acclimatization is absolutely required to overcome effects of the cable on trial counts, so that we can obtain robust measures of IP, slope, and transitivity. We have adjusted the Materials and methods section to be more explicit about this period. Testing then progressed over another 8 weeks, and as we now show in the supplemental material, we do not see any effects of inactivation on behavior even when we subdivide this period into early, middle, and late periods. And of course the late sessions were about a week prior to the PR testing. We hope this new information, now better described in the revision, mitigates the concern that testing prior to successful expression is the reason for the negative effects in the economic choice task.

2) In the progressive ratio task, the effect size of mOFC inhibition seems rather small (~2 trials), although consistent (Figure 7). What are comparable effect sizes reported in other manipulations? In reference to the PR task (Figure 7), where does the n=8 and F(1,10) come from?

Thank you for raising this issue as we feel we should add additional information for the behavior on the PR task. The procedures for the progressive ratio task are somewhat variable across labs in the amount of prior training and the schedules used. Results for the progressive ratio test can be reported as the final ratio reached – breaking point, or as total lever presses completed with both measures reported as the main measure in the literature. Many studies report the total lever presses, which is a finer measure of behavior – and presumably a more sensitive measure of comparison. We have added the total lever presses to our figure.

As for the degrees of freedom – As described in the paper, we ran four rats on the PR task, twice with the patent fiber and twice with the blocked fiber. For the analysis and in Figure 7A, we paired the first and second session and the third and fourth session (which were counterbalanced with both fiber-types) resulting in an n = 8. The degrees of freedom for the ANOVA are as follows: total df: 15 (4 rats x 2 fiber-types x 2 runs – 1); Fiber-type factor: 1 df; Rat factor (blocking factor): 3 df; Run factor (blocking factor): 1 df. Run being the first two sessions considered as a block and the latter two sessions considered as a block. The df for the error was 15 – (1+3+1) = 10.

3) It is not fully clear how disrupted economic choice would look like for specific behavioral metrics. What are the hypotheses regarding a disruption or changes in this task? For example, a change in IP would reflect a change in subjective value between choice options. Would inhibition of a value-computing area during parallel stimulus presentation expected to shift IP? Flatten IP function? Etc.

We agree that the proponents of this idea have not clearly specified what the effects of removing the “economic value generator” would be on behavior. We actually asked a number of neuroeconomists when we began this work and never got a straight answer. It seems to us that there are at least 3 principled potential effects: 1) The choices become completely random – this should result in a flattening of the slope 2) The specific or qualitative information of the value of the pellets is inaccessible – this should result in a shifting of the IP, presuming that valuing quantity is still intact. 3) The ability to appropriately determine the relative value across several goods is disrupted – this should disrupt the transitivity measure of three pellets. We test for all three effects, finding no evidence for any of them.

4) Has there been a demonstration that any manipulation can shift the IP?

Not to the best of our knowledge. The main problem in this regard is that obtaining robust measures of IP, slope and transitivity – which are required to meet the criteria for “economic choice” – require the subjects to do a certain number of trials. This precludes or makes very difficult most of the ways one might get at this question, since if you do not deliver the food (typically required in devaluation), there are not enough trials to derive reliable measures of the economic value.

5) Please clarify the pair contingencies in the Materials and methods section. Specifically, for 'transitivity' tests, how were the pellet flavor pairs chosen? What are the respective IPs for the chosen triplets for the six rats.

We have updated the Materials and methods section to be more explicit about this procedure, and we have included a supplementary table to go along with Figure 5 which contains all of the IPs. We also have updated Figure 5 to include an example of all data (6 sessions) which factor into one single measurement of transitive behavior. We hope this will make this figure more clear as to the nature of the data which enters into a single point. As for how we selected the pellet flavors: basically, we randomly chose which three pellets would be used for each of the separate repetitions of data collection for transitivity. We did make sure we sampled all of the pellets uniformly.

6) I did not fully understand the construction of the blocked-light patch cable.

We have adjusted the Materials and methods to better explain this.